**communications** engineering

# Mixed-mode in-memory computing: towards high-performance logic processing in a memristive crossbar array

Nan Du [1,2] ✉, Ilia Polian [3], Christopher Bengel[4], Kefeng Li[1,2], Ziang Chen [1,2], Xianyue Zhao [1,2], Uwe Hübner[2], Li-Wei Chen [3], Feng Liu[5], Massimiliano Di Ventra [6], Stephan Menzel [5] & Heidemarie Krüger[1,2]

In-memory computing is a promising alternative to traditional computer designs, as it helps overcome performance limits caused by the separation of memory and processing units. However, many current approaches struggle with unreliable device behavior, which affects data accuracy and efficiency. In this work, the authors present a new computing method that combines two types of operations — those based on electrical resistance and those based on voltage — within each memory cell. This design improves reliability and avoids the need for expensive current measurements. A new software tool also helps automate the design process, supporting highly parallel operations in dense two-dimensional memory arrays. The approach balances speed and space, making it practical for advanced computing tasks. Demonstrations include a digital adder and a key part of encryption module, showing both strong performance and accuracy. This work offers a new direction for reliable and efficient in-memory computing systems with real-world applications.

The increasing demands of data-intensive computing underscore the necessity for high-performance computing paradigms endowed with robust data processing capabilities. Today's von-Neumann computing architectures, based on mature complementary metal-oxide-semiconductor (CMOS) technology and transistors with latching characteristics for logic gates, face challenges in transistor scaling and the persistent memory wall issue. Despite advances in automation tools for logic circuitry design, these limitations necessitate a paradigm shift[1,2]. In-memory computing enabled by nanoscale memory technology[3,4] circumvents the memory-processor bottleneck by executing logic processing within the memory itself. Among enabling nanotechnologies[5], memristive crossbars stand out as particularly promising candidates due to their analog computing capability[6,7], low power cost and high scalability[8,9]. The key feature of a memristive element is its resistance change over time based on the current that flows though it (and some other microscopic degrees of freedom), integrating memory effect and latching characteristics in one, distinguishing it fundamentally from its transistor counterparts. In memory applications, data is stored as resistance (memristance) values in each cell, with Low Resistance State (LRS) representing '1' and High Resistance State (HRS) representing '0'.

For logic processing applications, known memristor-enabled computing paradigms are based on either stateful or nonstateful logic approaches. Stateful logic designs[10–13] capitalize on memristance states stored in individual cells as logic inputs and outputs, offering cost-effectiveness. However, their practical implementation in memristor crossbars faces substantial obstacles due to inherent reliability issues[14,15]. For instance, notable stateful logic designs[10,11] typically assess simultaneously multiple cells in a crossbar configuration for computing each logic gate. These designs confront challenges such as writing variations that reduce the programming window between logic '0' and '1'[16], as well as unintended programming of neighboring cells by programming voltages applied to target cell(s)[17,18]. Alternatively, the nonstateful approaches[19–21] leverage transient voltages or currents applied to (or sensed from) memristors as logic variables '0' and '1'. Though more resilient to variations, nonstateful designs lack universality (as clarified in Supplementary Information A). They require sense amplifiers controlled by dedicated peripherals to handle cascaded logic functions, leading to high power and latency cost in peripherals and compromising the in-memory computing benefits. Despite considerable efforts to advance memristive logic designs[12,13], the inherent trade-off between computing efficiency and accuracy persists in both approaches.

[1]Institute for Solid State Physics, Friedrich Schiller University Jena, Jena, Germany. [2]Leibniz Institute of Photonic Technology (IPHT), Jena, Germany. [3]Institute of Computer Engineering and Computer Architecture, University of Stuttgart, Stuttgart, Germany. [4]Institute of Materials in Electrical Engineering and Information Technology, RWTH Aachen University, Aachen, Germany. [5]Peter Grünberg Institut (PGI-7), Forschungszentrum Jülich, Jülich, Germany. [6]Department of Physics, University of California, San Diego, La Jolla, CA, USA. ✉e-mail: nan.du@leibniz-ipht.de

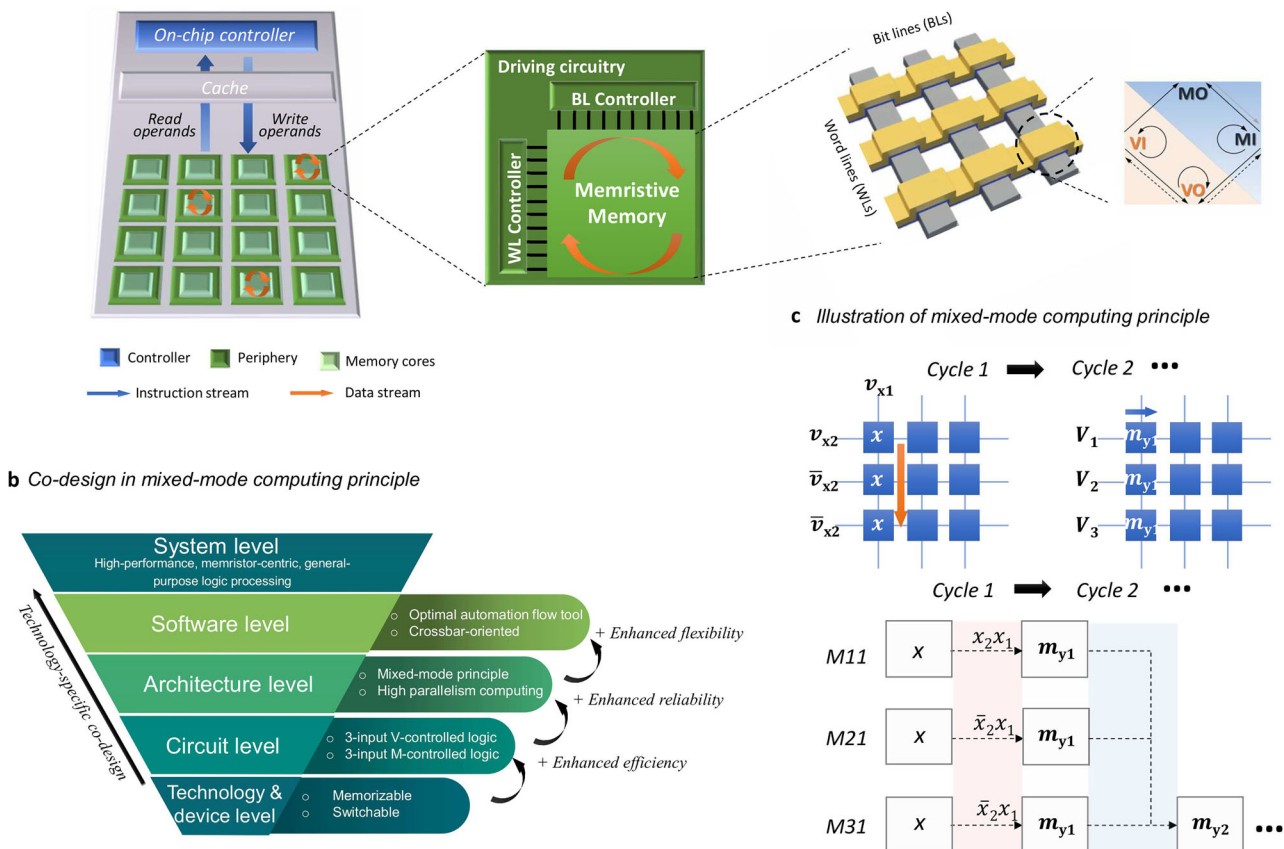

**a** *Mixed-mode computing paradigm*

**b** *Co-design in mixed-mode computing principle*

**c** *Illustration of mixed-mode computing principle*

**Fig. 1 | Mixed-mode in-memory computing and system-level co-design using memristive crossbars. a** Illustration of computing structures for in-memory computing with an inset on the right demonstrating working principle of mixed-mode computing, i.e. flexible integration of V- and M-mode logic operations in each memristive cell, including MI (memristance-input), MO (memristance-output), VI (voltage-input), and VO (voltage-output). **b** Co-design realized in system hierarchy using memristive crossbars for logic processing. **c** Example of mixed-mode computing using V and M modes in cycle 1 and 2, respectively. The computing is performed on memristive cells M11, M21 and M31 in one bit line in a memristive crossbar. In cycle 1 (V-mode), the voltage values $v_{x1}/v_{x2}/v_{x3}$ are applied to execute logic encoding based on the combinations of logic input variables $x_1/x_2/x_3$. In Cycle 2 (M-mode), fixed voltages $V_1/V_2/V_3$ are applied, and outputs are stored as resistance states $m_{y1}$ and $m_{y2}$ in corresponding memristive cells.

To tackle this issue, we propose an innovative approach: mixed-mode in-memory computing paradigm that maximize the use of memory effect and latching characteristics in each memristive element. Our approach offers effective solutions at the circuit, architecture, and software levels to address the reliability issues pertaining to memristive crossbars, while eliminating high cost current sensing during cascading.

In particular, we see that mapping computations to memristive crossbars differs substantially from classical logic synthesis in CMOS transistors. Simply adapting known tools[22], algorithms[23,24], or data structures[25–27] to memristive crossbars, as done in current solutions, is insufficient to fully exploit the high parallel computing capability of 2D crossbars. Therefore, we develop a dedicated crossbar-oriented automation tool for seamlessly integrating the voltage- and memristance-controlled logic operations. As a proof-of-principle, we have experimentally demonstrated the computation of an *N*-bit carry ripple adder and cryptographic S-Boxes. While striving for an optimal balance between processing latency and area, these demonstrations provide compelling evidence of its capability to solve arbitrary *n*-input functions with enhanced reliability in data processing.

## Results and discussion
### Mixed-mode in-memory computing framework
The key of innovative mixed-mode in-memory computing paradigm is to utilize each memristive cell for either M-mode, where information is

represented as resistance or memristance stored in the cell, or V-mode, where voltage applied to or current sensed from the same cell is used for this purpose, as illustrated in Fig. 1a. This allows firstly arbitrary combinations of stateful and nonstateful operations on user-defined subsets of the crossbar, maximizing their benefits while keeping their expected drawbacks in check. We propose here effective solutions at circuit-, architecture-, and software-level (Fig. 1b), for solving arbitrary complex logic functions with high performance while mitigating the intrinsic reliability issues in nanoscale memory devices.

At circuit- and architecture-level, we determine distinct four logic kernels serving as building blocks of mixed-mode computation: memristance-input (MI), memristance-output (MO), voltage-input (VI), and voltage-output (VO). MI or VI kernels employ memristance M or, respectively, voltage V as the logic input variable, while in MO or VO kernels, respectively, M or V serves as the logic output variable. Supplementary Information A highlights that all existing representative memristive logic designs in the literature can be systematically classified according to newly determined logic kernels. Remarkably, existing memristive logic designs in the same logic kernels show analogous benefits and constraints, affirming the validity of employing these distinct logic kernels for systematic categorization of memristive logic designs.

MI and VI kernels are representing the latching feature inherent in memristors, while MO and VO representing memory effect, enabling logic operations without changing their resistance states. Under the mixed-mode

principle, a computation is mapped to a series of logic operations in MI, MO, VI, and VO kernels, which can be flexibly interchanged and can use combinations of M- and V-modes at their inputs and outputs. Moreover, each cell in the array can be utilized for implementing single or multiple logic operations in each kernel, with MO facilitating low-cost cascading and storing the output memristance in the circuitry. For instance, in Fig. 1c, cells M11–M31 in a BL in crossbar undergo one cycle of logic operations in VI, followed by logic operations in MI during the subsequent cycle.

At software- and system-level, in order to maximize the considerable computing efficiency achievable through parallel operations across multiple memristive cells in a two-dimensional grid, we developed an approach called M³S (mixed-mode mapping and synthesis) that automatically synthesizes M-and V-mode operations to the cells for solving arbitrary logic function and determines the corresponding control signals on wordlines (WLs) and bitlines (BLs). While its algorithmic principle is generic, the tool used for demonstrations in this work balances between benefits and costs of the VO kernel (Supplementary Information B) by permitting only the initial values as logic input variables and restricting readout operations to the beginning of computation. Despite minimizing the power and latency costs in peripherals, it also mitigates the necessity for data copying or transmission within the crossbar, and further allows logic processing at arbitrary positions while ensuring necessary logic inputs for the VI kernel at any computing cycle.

## Highly efficient 3-input logic operations

For facilitating the mixed-mode logic cascading introduced in last section, we propose 3-input logic operations in each kernel, i.e. MI³, VI³, and VO³ operations, with MO as low cost logic cascading among them, as shown in Fig. 2a. All 3-input logic designs execute distinct logic functions within one single cycle. We determine $M_1$ as input and output cell. the memristance state $m_{x1}$ of $M_1$ prior to the logic operation and the memristance state $m_{y1}$ subsequent to it serves as one of the logic inputs and the logic output, respectively.

The MI³ operation from MI kernel exploits three parallel-connected memristive cells in one word line (WL) or bit line (BL) with memristance logic inputs $m_{x1-x3}$ that are stored in each device as a logic input variable. In contrast, the 3-input VI³ operation from VI operates exploiting deterministic writing operations on a single cell, utilizing the initial state $m_{x1}$ and the voltages applied to its device terminals $v_{x2}/v_{x3}$ as logic input variables. MI³ and VI³ operations implement the functions

$$(m_{y1}) = \text{MI}^3(m_{x1}, m_{x2}, m_{x3}), \tag{1}$$

$$(m_{y1}) = \text{VI}^3(m_{x1}, v_{x2}, v_{x3}). \tag{2}$$

Figure 2b, c illustrates the corresponding truth tables, respectively. The logic functions are implemented on Au/BiFeO₃/Pt/Ti memristors (Fig. 2d). The output $m_{y1}$ can subsequently function as logic input for both cascaded VI³ and MI³, thereby greatly facilitating logic cascading during automation design. Intuitively, the 3-input MI³ and VI³ operations can be considered as extended 2-input memristor-aided logic (MAGIC)[11] and complementary resistive switching (CRS, without readout) logic designs[20], which encounter inevitable high error rates and universality issues, respectively, as detailed in Supplementary Information A.

The mixed-mode computing framework offers solutions enabling the practical logic processing by exploiting MI³ and VI³ operations. Traditional MI kernels alone suffer from longer cycles and higher error rates due to the stochastic variability of memristive technologies[14], while VI kernels alone are not universal and require readout VO to realize arbitrary functions (Supplementary Information A). These limitations are overcome with the proposed mixed-mode approach, which integrates VI³ with MI³ operations to achieve universality without the need for VO, resulting in shorter processing cycles and enhanced resilience by offloading part of the functionality to the more reliable VI³ operations. Additionally, the mixed-mode computing principle addresses common issues in stateful logic designs such as

"input drift" and "partial switching" (Supplementary Information C), enabling practically cascadable MI³ operations. By setting the programming bias to $V_{in}$ = 5.3 V, we ensure correct transitions from LRS to HRS in $M_1$ with input combinations '101', '110', and '111', while preventing changes in memristance values in the input cells $M_{2-3}$. As expected, this $V_{in}$ results in a compromised LRS in $M_1$ for the '100' input combination (27.2 MΩ compared to the initialized 1.1 MΩ). As next, the M³S automation tool prioritizes using the output cell $M_1$ as an input cell in cascaded MI³ (or VI³) operations, allowing the compromised LRS in $M_1$ to be re-switched back to logic '1' through a positive $V_{in}$ applied to the top electrode of $M_1$ while correctly computing the cascaded MI³ output. These methods enable robust and deeply cascadable MI³ operations in practical implementations.

Figure 2e, f demonstrates the experimental results on MI³ and VI³ operations recorded from the fabricated BiFeO₃-based self-rectifying crossbar array (extended experimental results in Supplementary Information C). The Experimental Section depicts the fabrication and characterization details of the self-rectifying crossbar array based on BiFeO₃ memristors, with Au top electrodes (bottom electrodes) interconnecting as BLs (WLs). It is important to emphasize that the proposed mixed-mode approach is compatible with various nanotechnologies, including passive crossbar arrays and 1-transistor-1-resistor (1T1R) arrays (Supplementary Information D). Supplementary Information B presents the designed 3-input VO³ operation using different types of BiFeO₃ memristors in a comparative manner, effectively illustrating the technology dependency in the VO kernel. With MI operations at our disposal, VO is not necessary for universality.

## M³S: mixed-mode mapping and synthesis tool

Mixed-mode in-memory computing, with its V-mode and M-mode operations running simultaneously on a memristive crossbar, is so distinct from a conventional gate-based CMOS circuit that a mere adaptation of an existing synthesis tool is insufficient. The key to achieve an optimal solution with minimized requirements for cell and cycle numbers in in-memory computing, is to align the physical data location (mapping) following each gate operation in every computing cycle (synthesis) for enabling highly parallel computing in a two-dimensional memristive crossbar given its sequential computing nature. We developed the M³S tool, which concurrently addresses both mapping and synthesis tasks by formulating them as a Boolean satisfiability formula in conjunctive normal form, such that the solution (satisfying assignment) of this formula gives, for each crossbar location and each cycle, the V-mode or M-mode operations executed.

While M³S tool is capable of generating conjunctive normal forms that flexibly combine arbitrary V-mode and M-mode operations, the proof-of-principle demonstrations in this study utilize V-mode phase followed by M-mode phase, satisfying the computational requirements for the desired logic functions. Taking VI³ operation in V-mode as an example (Fig. 2c), we assume that the peripherals can apply input voltages $v_{x2-x3}$ according to functions from the list of literals $\{l_1 = \text{const} - 0, l_2 = \text{const} - 1, l_3 = x_1, l_4 = \bar{x}_1, l_3 = x_2, l_4 = \bar{x}_2, \ldots\}$. Therefore, the Boolean satisfiability formula determines for each crossbar location $M_{ab}$ $(a, b)$ and each cycle $c$ the variables $g^{TE}_{a,b,j,c}$ and $g^{BE}_{a,b,k,c}$ for voltage inputs $v_{x2} = l_j$ to top electrode and $v_{x3} = l_k$ to bottom electrode while executing each VI³. For instance, the VI³ operation on the crossbar cell $M_{78}$ in cycle 2 can be transformed into conjunctive normal form as shown in Eq. (3):

$$\bigwedge_{\substack{1 \le j, k \le 2n+2 \\ 1 \le q \le 2^n}} \left( (g^{TE}_{7,8,j,2} \wedge g^{BE}_{7,8,k,2}) \to \left( v_{7,8,2,q} \equiv \text{VI}^3(v_{7,8,1,q}, l_{j,q}, l_{k,q}) \right) \right). \tag{3}$$

Here, the expression $g^{TE}_{7,8,j,2} \wedge g^{BE}_{7,8,k,2}$ on the left side of the implication (→ ") defines the mapping constraints, including cell location $(a, b)$, cycle number $c$, and input bias-related literals $j/k$. If the solution to the Boolean satisfiability formula results in $g^{TE}_{7,8,1,2} = 1$ and $g^{BE}_{7,8,6,2} = 1$, it indicates that during cycle 2, the top electrode and bottom electrode of M78 are driven by

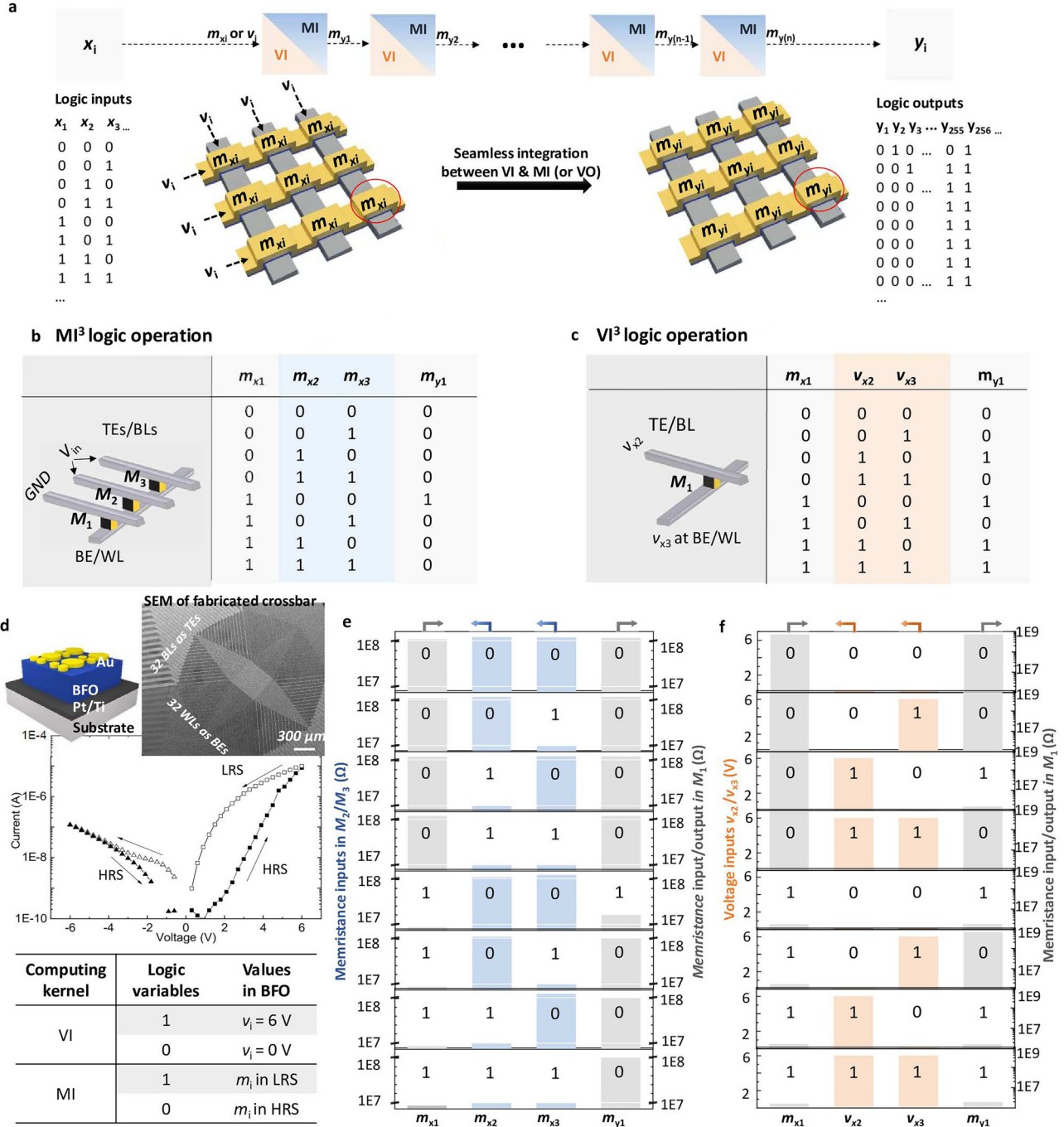

**Fig. 2 | Experimental demonstration of 3-input MI³ and VI³ operations by exploiting BiFeO₃ memristive cells.** **a** Demonstration of mixed-mode computing principle, transferring logic inputs $x_i$ into $y_i$ through seamless integration between VI and MI kernels (VO kernel is strictly excluded during cascading due to high cost as depicted in Supplementary Information B). Inset below illustrates the cell states in self-rectifying passive memristive crossbar structures before and after processing mixed-mode computing. Illustration of truth tables and circuit designs for **b** MI³ and **c** VI³ logic operations. GND indicates the grounded electrode. **d** Schematics and current-voltage ($I$-$V$) characteristics of BiFeO₃ memristive devices applied in this work with insets showing the SEM of fabricated passive crossbar and the definition of logic variables. The accompanying table summarizes the applied bias voltages $v_i$

for logic `1' and `0' in the VI computing kernel, and defines the corresponding memristance states $m_i$ in the MI kernel, where logic `1' and `0' are represented by low-resistance state (LRS) and high-resistance state (HRS), respectively. Experimental demonstration of 3-input **e** MI³ ($V_{in}$ = 5.3 V) and **f** VI³ (input logic `1' as $v_W$ = 6 V) for each input combination in the truth table. The memristance inputs $m_{x2}/m_{x3}$ (marked in blue) in MI³ and the voltage logic inputs $v_{x2}/v_{x3}$ (marked in red) in VI³ are demonstrated on left y axis, while the memristance input $m_{x1}$ and output $m_{y1}$ states (marked in gray) are shown on the right y-axis. State verification of these results is conducted through memory validation MO by applying a readout step at the end with a bias of 2 V to the top electrode.

---

$l_1$ = const-0 and $l_6 = \overline{x_2}$. Conversely, the right side of the implication ($\rightarrow$ ") with $v_{7,8,2,q} \equiv$ VI³($v_{7,8,1,q}$, $l_{j,q}$, $l_{k,q}$) specifies the conditions for the VI³ operation in cycle 2. This encompasses the synthesis conditions, such as the definitions of the logic functions VI³ or MI³ and the operated $q$-th entry.

Hence, based on the memristance input variable $v_{7,8,1,q}$ in M78 during cycle 1, the output variable $v_{7,8,2,q}$ in cycle 2 is set to 1 if the $q$-th entry of the VI³ truth table is 1, and 0 otherwise. Detailed descriptions of the M³S design in V- and M-mode can be found in Supplementary Information F. The parallel

computing strategy, which cascades across multiple cells in the 2D crossbar array, is ensured by transforming all $VI^3$ and $MI^3$ operations into conjunctive normal forms for each cycle. Additional constraints ensure that all cells sharing the same WL (bottom electrode) and BL (top electrode) as $Mab$ adhere to the same input-related literals $lj$. $M^3S$ computes all coordinates simultaneously, achieving an optimal solution with minimized crossbar size and cycle count while adhering to the robust logic cascading conditions outlined previously.

Unlike the previous work[28], which relies on classical synthesis tools like ABC and CPLEX, we build directly from Boolean functions, and employ at least two modes (MI and VI) rather than the single MAGIC logic (MI mode). Additionally, our approach integrates synthesis and mapping from the outset, simultaneously generating a gate netlist while considering the physical locations of memory cells and the parallelism inherent in crossbar architectures. This co-design methodology ensures that synthesis directly exploits the crossbar's computing potential, leading to superior efficiency and performance and offering optimality guarantees.

## Experimental proof-of-principle demonstrations

We demonstrate the advantages of using the mixed-mode computing paradigm on two examples: $N$-bit carry-ripple adders and a small-scale (4-bit) version of the S-Box from the Advanced Encryption Algorithm (AES)[29]. An $N$-bit carry-ripple adder is an example of modular design that consists of identical cells (full adders); there are numerous prior implementations of adders that can serve for comparison. An S-Box, which provides the necessary nonlinearity during encryption, is an algebraically complex construction that is too large to be optimized in a handcrafted manner. Note that security aspects of the S-Box (e.g., its vulnerability to side-channel attacks) are not in scope of this work.

Utilizing the automation tool $M^3S$, a control sequence has been generated for implementing the full adder $N$-bit carry-ripple adder as depicted in Fig. 3a, utilizing BiFeO$_3$-based crossbar architectures. The control sequence illustrated in Fig. 3b maps arbitrary combinations of logic inputs $c_i$, $a_i$, and $b_i$ (represented as $x_{1-3_i}$, respectively) onto the desired logic outputs $c_{i+1}$ and $s_i$ (represented as $y_{1-2_i}$) through $VI^3$ and $MI^3$ operations. According to carry-ripple adder's three input $x_{1-3_i}$, one of the eight values (literals) $x_{1_i}$, $\bar{x}_{1_i}$, $x_{2_i}$, $\bar{x}_{2_i}$, $x_{3_i}$, $\bar{x}_{3_i}$, const-0 and const-1 can be applied to either of the memristive cell's two electrodes, with the first connected to the top electrode and the second to the bottom electrode of each cell, which correspond to BLs and WLs in a crossbar array (Fig. 3b). Starting with the unknown state $x$, each $VI^3$ operation computes $m_{y_i}$ and stores it directly as a memristive state, which can serve as input to subsequent $VI^3$ and $MI^3$ operations. The control sequence depicted in Fig. 3b contains 4 rows, each representing a cell required for computing a 1-bit carry-ripple adder, with 1 additional cell dedicated to storing the output-carry bit $c_{i+1}$). Furthermore, the 5 numbered columns signify the requisite five cycles for executing the 1-bit carry-ripple adder: 3 cycles involve $VI^3$ operations concurrently across all cells (highlighted in red), while 2 subsequent cycles involve $MI^3$ operations (highlighted in blue). Extending such design to an $N$-bit adder reveals that only the variation-resilient $VI^3$ operation for computing the output carry bit $c_{i+1}$ in cycle 2 necessitates operations for each bit in $a_i$ and $b_i$, resulting in a total of $N + 4$ cycles and $4N + 1$ cells. As aforementioned, experimental processing typically incurs high error rates with the MI kernel, while the VI kernel remains relatively error-free. It is notable that regardless of $N$'s size, an $N$-bit adder utilizing this design contains only 2 cycles of parallel $MI^3$ operations, implying consistent error rates across varying $N$. Due to space constraints, herewith we demonstrate the experimental results of a 4-bit carry-ripple adder.

Supplementary Table S4 in the Supplementary Information I shows that all representative $N$-bit adder designs[12,23,30–42] are based on either MI or VI kernels. Our mixed-mode approach, being the first to exploit both kernels without current sensing during cascading, showcases an optimal balance between cell and cycle numbers. The only design with a better Area-Delay-Product lacks crossbar compatibility or necessitates substantial structural modifications for practical implementation. We highlight that our adder has been experimentally demonstrated in full, while many previous papers have characterize individual gates experimentally and then plug the results of such measurements into a simulation. Moreover, while our design eliminates current sensing during cascading, a separate set of results for adders incorporating the $VO^3$ operation is found in Supplementary Information G, along with comparable adders from literature[43–48]. Supplementary Information H summarizes the energy costs of adders with and without readout.

The block diagram of the 4-bit S-Box and its truth table are shown in Fig. 4a. Note that its input $x_1$-$x_4$ and output $y_1$-$y_4$ are encoded using 4 bits. Its control sequence found by $M^3S$ and shown in Fig. 4b, has the minimum possible number of 4 $MI^3$ operations, equal to 4 outputs and 12 cells (the 4 MI cycles are shown in parallel in Fig. 4b to save space). Figure 4c demonstrates the $VI^3$ and $MI^3$ operations of 4-bit S-Box with arbitrary inputs, where the bitwise parallel operations are performed with shared bottom electrodes in one BL in cycles 1–5 in individual cells, and the $MI^3$ operations are executed sequentially in cells M1-M3, M4-M6, M7-M9, and M10-M12, respectively. Compared with 4-bit S-Box exploiting 12 memristive cells by using mixed-mode principle, the prior CMOS implementation in ref. 49 obtained by Altera synthesis software had roughly 106 transistors (8 two-input gates, 8 three-input gates, 2 four-input gates, 1 five-input gate), and the CMOS circuit using as the basis for the memristive implementation in ref. 50 had 32 two-input gates or roughly 128 transistors. The experimental results of 4-bit Sbox on a passive crossbar based on BiFeO$_3$ memristive crossbar with inputs 1010 and 1111 by using the control sequence are presented in Fig. 4b.

This analysis does not consider the effort for peripherals that supply the top electrode and bottom electrode values. In comparison to peripherals required for other memristive implementations, the mixed-mode VI-MI operation incurs the same effort for reading but might add complexity for input-dependent writing biases. While we do not explicitly examine the endurance or retention properties of the memristors in this work, we are encouraging the synthesis procedure to use the variation-resilient VI kernel as much as possible, while resorting to the MI kernel only when this is required for universality. This consideration supports the flexibility of our synthesis approach: once M-mode operations will get more reliable in the future, the designer can simply rerun $M^3S$ with more clock cycles allocated to them.

## Conclusions

In this work, we propose a mixed-mode in-memory computing paradigm by seamlessly integrating V-mode and M-mode operations within each physical memristive cell. This approach combines the strengths of diverse gate processing (V-mode) and universality (M-mode) while mitigating the impact of device variations inherent in nanodevices. According to our review of current research, this approach represents the first practical implementation enabling complex functions in memristive crossbars, while eliminating the need for high-cost current sensing in peripherals during cascading.

Moreover, the first-of-its-kind memristor crossbar-oriented mapping and synthesis ($M^3S$) automation tool is developed, enabling flexible combinations of V-mode and M-mode operations for implementing arbitrary logic functions by carrying out synthesis and mapping tasks in one. $M^3S$ achieves an optimal balance between latency and area, reducing the typical trade-offs of conventional in-memory computing and enhancing power efficiency. Our experimental demonstrations include a best-in-class $N$-bit carry ripple adder and the first implementation of complex S-Box circuitry with 12 memristive cells. This showcases not only a significant reduction in device count—by an order of magnitude compared to conventional CMOS technology—but also highlights the enhanced robustness and potential of the mixed-mode computing paradigm.

The proposed mixed-mode computing is adaptable and transferable across various nanoscale memory technologies that feature nonvolatile and latching characteristics. In this work, while utilizing passive 1R crossbar array for its potential high density, the future work can extend to the 1T1R crossbar array which allows additional inputs to transistors to

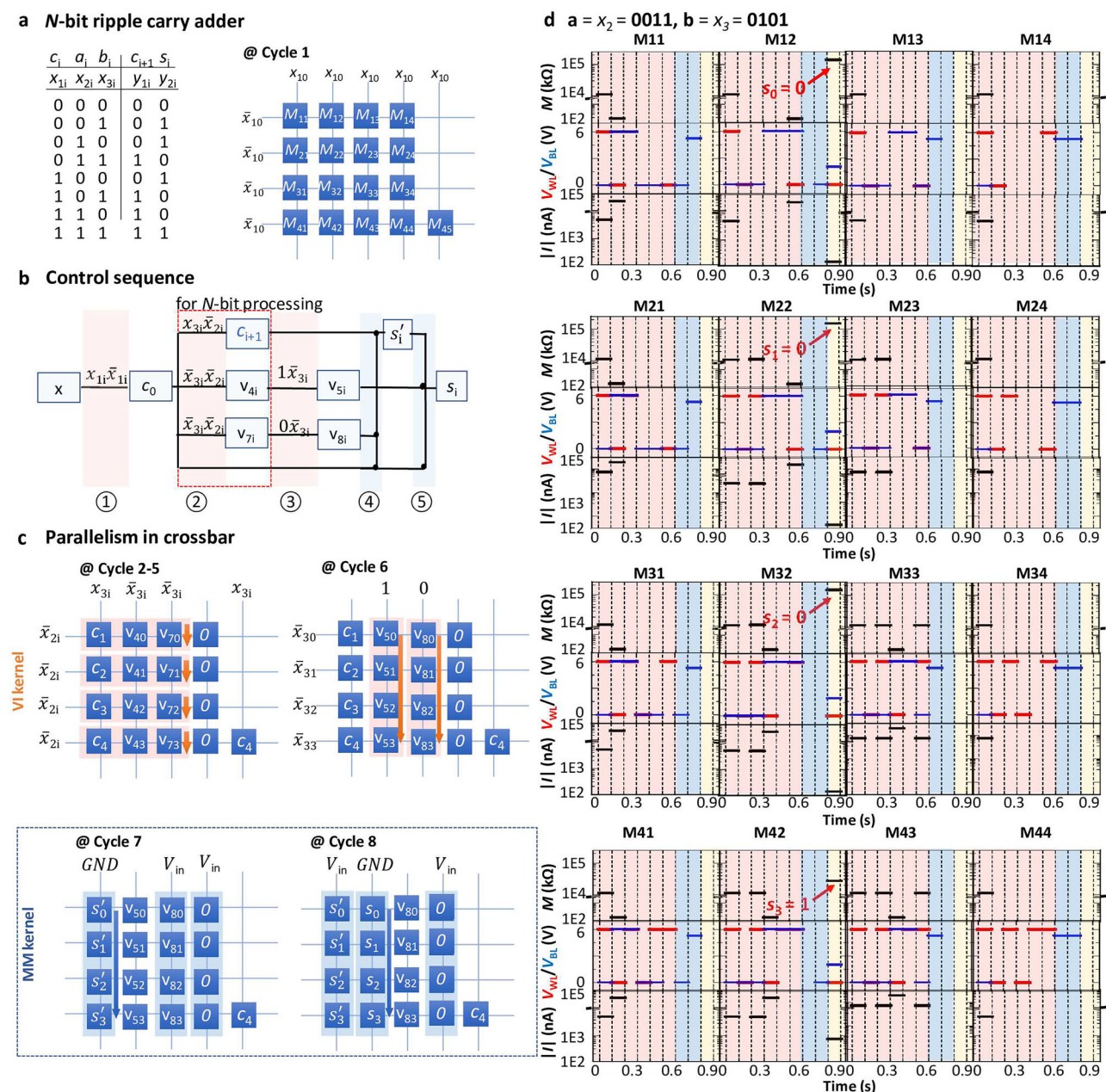

**Fig. 3 | Experimental implementation of *N*-bit carry-ripple adder by exploiting automation tool M³S. a** Demonstration of the truth table of *N*-bit carry-ripple adder and the exploited crossbar structure. **b** Control sequence of *N*-bit carry-ripple adder using BiFeO₃ memristive crossbar synthesized exploiting M³S. Each VI³ operation is noted as *v* in the diagram. **c** Illustration of logic operations for 4-bit carry-ripple adder with arbitrary inputs, which requires in total 8 cycles, i.e., 6 of VI³ and 2 of MI³ operations. **d** Experimental results of 4-bit carry-ripple adder, including memristance of each cell $M_i$, voltage on WL to shared bottom electrode $V_{WL}$, voltage on BL to top electrode $V_{BL}$, and the absolute values of current across each cell $|I_i|$ of 4-bit carry-ripple adder with inputs $a = 0011$, $b = 0101$.

serve as logic inputs, thus facilitating advanced logic operations in V and M kernels within the mixed-mode computing framework. Furthermore, the inherent analog computing capabilities of memristive devices enabling the design of ternary and multi-nary logic operations within the mixed-mode computing paradigm not only expands the computational horizons into mixed-nary computing but also heralds a new era of unparalleled efficiency in computing paradigms, promising unmatched computational power and efficacy.

## Methods
### Passive memristive crossbar fabrication
To fabricate the passive memristive crossbar array based on BiFeO₃ thin film series, i.e. BiFeO₃, BiFeTiO₃, and BiFeTiO₃/BiFeO₃ thin films,

the corresponding polycrystalline BiFeO₃ film series are deposited by pulsed laser deposition at 650 °C in oxygen ambient, respectively. Upon the Pt/Ti-bottom electrode patterned by photolithography and ion beam etching, for fabricating the BiFeTiO₃ or BiFeO₃ memristive crossbar arrays, the Ti doped BiFeO₃ thin film or undoped BiFeO₃ thin film are deposited on structured Pt (100 nm)/Sapphire and Pt (100 nm)/Ti (50 nm)/Sapphire substrates, respectively. Both BiFeTiO₃ and BiFeO₃ thin films possess rhombohedrally distorted perovskite structure (R3c space group) and a nominal thickness of 550 nm. In BiFeO₃ thin film, the ambient hitting during deposition provokes the substitutional incorporation of Ti donors into the BiFeO₃ lattice, which constructs a rectifying/non-rectifying contact with flexible barrier height near to top/bottom electrode region, whereas, in BiFeTiO₃

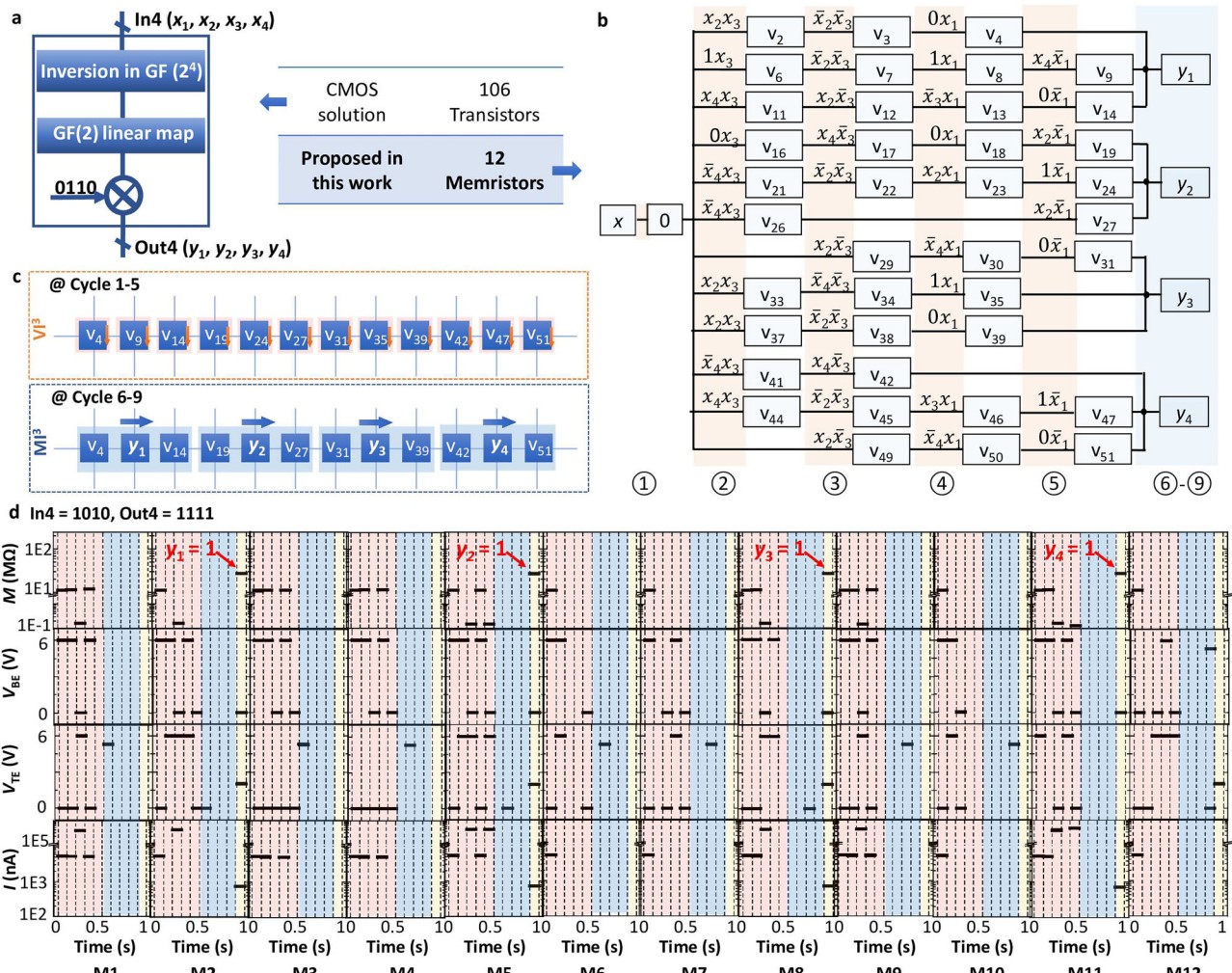

**Fig. 4 | Demonstration and implementation of 4-bit S-Box based on automatic synthesis algorithm. a** Block diagram and comparison of cell numbers between proposed solution and CMOS implementation from ref. 49. GF denotes Galois Field. **b** Control sequence provided by M³S using BiFeO₃ based memristive crossbar. Each VI³ operation is noted as v in the diagram. The red and blue shadowed cycles are VI and MI cycles, respectively. **c** Demonstration of sequential operations with arbitrary inputs. **d** Experimental results with input $A_{hex}$ ($x_{1-4}$ = 1010) and output $F_{hex}$ ($y_{1-4}$ = 1111). The results of the memristance of each cell $M_i$, the voltage on WL to shared bottom electrode $V_{WL}$, the voltage on BL to top electrode $V_{BLi}$, and the absolute values of the current $|I_i|$ across each cell during operation are demonstrated.

thin film the Ti content of nominal 1 at% promotes the modulation of the flexible barrier in the top electrode region. The subsequent deposition of BiFeTiO₃ (100 nm) and BiFeO₃ films (500 nm) on structured Pt/Sapphire substrate sets up the BiFeTiO₃/BiFeO₃ bilayer structure in BiFeTiO₃/BiFeO₃ memristive crossbar array with the flexible Schottky-like barriers formed at the top and bottom interfaces for the bilayer structure. The Au top electrode with thickness of 180 nm was evaporated and patterned by photolithography followed by a lift-off. As an example, the SEM image of 32 × 32 BiFeO₃ based crossbar array (with a junction area of each cell as 20 × 20 μm² and a pitch of 25 μm) is demonstrated in Fig. 2a.

## Electrical characterization

All the experimental electrical measurement illustrated in this work were recorded using a probe station and a Keithley source meter 2400, which is controlled by a home-made PCB board for selecting and applying biases to WLs and BLs through LabVIEW program.

## Data availability

The data that support the plots within this paper and other findings of this study are available from the corresponding author upon reasonable request.

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

## Acknowledgements

This work is supported by the German Research Foundation (DFG) Projects MemDPU (Grant Nrs. DU 1896/3-1 and ME 4612/1-1). I.P., N.D., L.C., Z.C. and X.Z. acknowledge the funding support by DFG project MemCrypto (GrantNr. DU 1896/2-2 and Grant Nr. PO 1220/15-2). S.M., F.L. and C.B., acknowledge the funding support in part by the Federal Ministry of Education and Research (BMBF, Germany) through the project NEUROTEC II with the grant numbers 16ME0398K and 16ME0399. M. D. acknowledges support from the National Science Foundation under Grant No. ECCS-2229880.

## Author contributions

N.D. and S.M. conceived the main conceptual idea and computational framework of mixed-mode computing. I.P. designed the crossbar-oriented automation tool $M^3S$, and optimized control sequence for $N$-bit carry-ripple adder and 4-input Sbox. N.D. and I.P. wrote the manuscript with the support from S.M. C.B. provides the energy evaluation of $N$-bit carry-ripple adder implementations. K. L., Z.C. and X.Z carried out the experimental work and analysed the data. L.C., and F.L. aided in interpreting the results. U.H. fabricated the physical crossbar based on $BiFeO_3$ memristors and performed SEM analysis. H.K. initiates and supervised the work. M.D. supported data interpretation and provided critical feedback, especially in computing reliability in memory-oriented logic design considered in this work. All coauthors discussed the results and implications at all states and contributed to the improvement of the manuscript text.

## Funding

## Competing interests

The authors declare no competing interests.
