## [Transparent Peer Review file · Communications Engineering]

Mixed-Mode In-Memory Computing: Towards High-Performance Logic Processing in Memristive Crossbar Array

Corresponding Author: Dr Nan Du

Version 0:

Reviewer comments:

Reviewer #1

(Remarks to the Author)

It seems that the key contributions are threefold. First, it is a "hybrid" memristor Boolean logic that combines previously suggested flavors of memristor logic, most notably including material implication and MAGIC logic. Second, it is a design automation tool for mapping combinational logic circuits to the proposed ones. And third, it is an experimental demonstration of a 4-bit ripple carry adder and 4-bit input/output S-Box, a basic module used in cryptography.

The results are quite heroic and impressive. However, I am missing the big picture and how the proposed work can advance the field of computing. Would there be any advantages to implementing adders and S-box modules with the proposed technology compared to the CMOS one, and if so, what are they?

For example, on the device level, the demonstrated delay, energy, and energy-delay product for minimum-width NMOS transistors are below 1 ps, 1 aJ, and 10^{-31} Js, respectively, for 10nm node technology. The write speed, energy, and their product are many orders of magnitude worse for memristors, even when using optimistic projections. Hence, it seems that the performance of memristor logic, which involves switching resistive states for logic operation, is fundamentally inferior to CMOS logic, and that neglects another serious issue of the limited switching endurance of memristors.

So, I would like to see the discussion of specific practical applications that would be enabled by or for which the proposed logic implementation would be superior to the conventional (CMOS) technology.

My other major concern is that the paper is challenging to read and contains many significant typos (e.g., in panel labeling in Figure 3 and how they are referred to in the text).

Reviewer #2

(Remarks to the Author)

Comment: This manuscript proposed a 3-input logic operation based on the memristors resistive state or operating voltage as input variables through a synergistic design at the device-system level. A mixed-mode in-memory computing method is constructed accordingly. The proposed computational method is experimentally demonstrated using a fabricated self-rectifying memristor array, and its application potential in specific scenarios is evaluated through a ripple carry adder and an S-BOX encryption task. However, the following issues should be addressed before the paper is accepted for publication.

Methodological aspects:

1. The authors need to further explain the differences between the proposed 3-input method and the mentioned two-input method, aside from adding device resistance as an input variable. How does this addition improve the efficiency and accuracy of logical computations? Furthermore, the authors should clearly explain why this 3-input method is more conducive to logic cascading compared to the two-input method.
2. The authors should provide a clearer explanation of the working principle of the mapping synthesis tool. The description of equation (3) is not easily understood. It might be helpful to include pseudo-code or a program flowchart. If possible, the authors should also explain how the logic synthesis tool is used to synthesize specific application examples in the subsequent application section.
3. In terms of application examples, the authors need to present results in a clearer way. For example, in the application of the ripple carry adder, it would be better to indicate which data are being added together and what result is obtained. The current plotting figures are not convenient for readers to understand.
4. Can the authors provide a quantitative performance improvement for the application examples? For instance, compared to

CMOS and existing memristor logic computation methods, how much improvement in computational energy efficiency and speed can be achieved at different bit widths?

Experimental aspects:

5. The device data needs to be supplemented. Since the authors rely on the device's resistive state switching to perform logic operations, the characteristics of the device's resistive state retention, endurance, variation, switching ratio, etc., will all impact the performance of logical computations. Therefore, the authors should supplement this data.
6. It would be beneficial for the authors to analyze the impact of non-ideal effects of the device on the proposed logic computation method. Which factor has the greatest impact? What effect does it have on the logic computation results? Compared to existing computation methods, does the proposed 3-input method offer better robustness?
7. The authors used SRM devices for experimental verification. Does the use of SRM devices, compared to bipolar devices or 1T1R devices, contribute to enhancing the performance of the 3-input logic computation method? In what aspects specifically?
8. The authors mentioned that this method can be transferred to different devices such as STT-MRAM. Does the performance of the 3-input logic method vary on different devices? Which one do the authors most recommend as the computing medium for this method, and why?

Grammar and writing aspects:

9. The authors should reduce the use of abbreviations. The extensive use of abbreviations and the confusion of various concepts make the paper difficult to understand and obscure the presentation of the paper's highlights.
10. Can equation (3) be broken down into a set of equations to explain the working principle of the synthesis tool? The current equation (3) is difficult to grasp quickly.
11. Page 5, Line 18 contains a spelling error.

Reviewer #3

(Remarks to the Author)

The paper proposes a novel mixed-mode paradigm for performing logic processing using memristor crossbar arrays. It improves upon state-of-the-art approaches by providing a framework in which both memristance-controlled and voltage-controlled logic processing units can be efficiently integrated, thereby overcoming issues arising out of implementing only one. They do so by proposing extended and improved versions of each and developing a custom automation tool that can synthesize and map arbitrary logic functions into optimal combinations of voltage and memristance-controlled processing units while being aware of physical constraints. Finally, the proposed ideas are validated through experimental demonstrations of two exemplary circuit blocks on memristor crossbar arrays.

While the proposed ideas are novel, comparison with prior work fair and experimental validations impressive, the paper could significantly benefit from better figures (especially the conceptual ones, such that they are easier to interpret by just looking at the figure itself without having to refer to much of the related explanation in main text or caption) and further explanation/elaboration of some of the key concepts/techniques presented. While the current version of the manuscript can already be impactful, resolving the points mentioned below may help make the key concepts and techniques presented more comprehensible and easier for other researchers to reproduce this work.

1. Although the truth tables of the proposed VI3 and MI3 logic processing units are provided there is a paucity of details regarding the standard logic gate emulation capabilities of these proposed units. The manuscript could benefit from elucidation on the logic gates that the MI3 & VI3 can natively implement. This can be done by (a) writing explicitly the Boolean function that these proposed units implement (the Boolean function + digital logic schematic corresponding to their truth tables). Then (b) providing details on how these natively implementable functions can be used to build more complex or a variety of other Boolean functions like AND, OR, XOR, XNOR etc (with 2,3 and more inputs).
2. Although successful experimental demonstration of n-input CRA and 4-bit S-box is shown, it would be great if the steps, techniques and concepts required to map such a circuit using the proposed mixed-mode computing paradigm to the crossbars is elucidated. Authors do provide details related to this in table in Fig. S6 and sub parts of Fig. 3, but the flow is difficult to comprehend. This could potentially be resolved for instance by taking a 1-bit full adder and (a) first describing its schematic using standard logic gates, (b) then its schematic using the natively implementable logic gates provided by the VI3+MI3 hybrid circuit and then (c) schematic of their physical implementation on a crossbar, where each new cycle could be described in a new crossbar diagram. Another important aspect that could be elaborated upon through text/explanation and/or figure/diagram is (d) how would the number of cycles, crosspoint devices required change if the 1-bit full adder is entirely implemented with VI3 or MI3. This would make the message that mixed-mode is better than implementing entirely voltage or memristance-mode more convincing.
3. The SAT formula that is optimized in the mixed mode mapping and synthesis tool could benefit from further elaboration. One suggestion is to take a simple Boolean function(it could be the 1-bit full adder or something simpler) and write down the exact SAT formula (detailing all the clauses) that would be optimized to find the corresponding mixed-mode implementation.
4. In the first point of the list of limitations that stateful logic faces (in page 4, Section A of SI), authors claim that in such stateful implementations only one type of switching is performed, either SET or RESET. However, even in the proposed MI3 kernel only a LRS to HRS transition takes place (if any). So, it is not clear to how introducing the MI3 kernel alleviates this limitation. On the other hand, the VI3 kernel does allow both types of transition (LRS to HRS as well as HRS to LRS). So, is it through the mixed-mode utilization of MI3 & VI3 kernels (not just the sole usage of MI3) that this issue is alleviated?
5. The authors describe the issue of partial switching/compromised state and unwanted drift in both input and output memristances in standard stateful logic implementations and allude to the fact that such issues are reduced by techniques proposed in this work (in Section C of SI). The only thing that does seem to fundamentally reduce such program disturb

errors is the idea of using hybrid VI + MI logic units. The maximum error levels caused by cascading pure MI style logic units can be reduced by intermittently introducing VI style logic units in between. However, aside from that, the issue of program disturb still fundamentally exists in the MI3 kernel due to the voltage divider circuit. Authors mention having found an optimal bias of 5.3V that reduces drift in input memristance. However, finding such optimal biases for voltage divider circuits is an obvious pre-processing step that need to be undertaken for other memristor devices as well. Moreover, such biases are prone to device-to-device variation effects. Therefore, optimal biasing is necessary but not a fundamentally fool-proof way of reducing such drifts. The idea that disturbed states of output memristors in a previous stage of MI3 kernel can be fixed by making it the input memristor in a subsequent stage, does not seem to be intentional, rather a by-product of the cascading process. Wouldn't such a trick be implemented by default when memristance-controlled logic style (either using stateful logic styles in literature like MAGIC or the newly proposed MI3 kernel) are cascaded? Can the authors comment on this? And if it is true, authors should be careful to not imply that such tricks are fundamental and non-trivial contributions of this paper that helps reduce program disturb errors in newly proposed MI3 kernels.

6. In page 4, Section A of SI, authors claim the following: "This advancement not only broadens the input variables from 2 to 3, facilitating the cascading process between MI and VI kernels as well as within the MI kernel". It is not clear how and to what extent this advancement is important. Firstly, can the authors quantitatively state the benefits arising out of this additional input in the new 3-input version compared to 2-input versions in literature? Secondly, in what ways is the MI3 kernel responsible for facilitating the cascading between MI and VI kernels? For instance, envision a mixed-mode computing paradigm that comprises of the proposed VI3 kernel (for the voltage-controlled logic unit) and the older 2-input MAGIC-like kernel (for the memristance-controlled logic unit). Wouldn't a cascading still be possible? If so, then can the authors comment on what is the true benefit of the new MI3 kernel?

7. In Section B of SI, the authors claim that VO style kernels can be used to reduce data copying and transmission operations in mixed-mode computation. Authors utilize such kernels only at the beginning of logic processing to facilitate retrieval of input data from memory to input VI kernels. However, will it not be useful also in intermediate stages? For instance, when MI3 kernels are cascaded, it may be possible that the input memristors of a later stage is not physically present at the same crosspoint location as the output memristor of a previous stage that generates that. In such cases, one would have to incur copying/transmission cycles. Can the VO3 kernel not be useful to efficiently relay such physically disjoint memristance values across crossbars?

8. Fig. S3 is not easy to read. Please edit the caption and/or figure to make the following fact more easily interpretable: the state S in (a) is the same as mx1 in (d-f) and that it indicates the physical configuration of the memristor after a write pulse is applied .

9. Type: "demonstrators" to "demonstrations" in first paragraph of page 5 of main text.

10. The authors claim "To the best of our knowledge, this is the first approach to integrated synthesis and mapping of Boolean functions on a crossbar, and also the first to offer optimality guarantees". Is this statement accurate even in light of other works like "SIMPLE MAGIC: Synthesis and In-memory Mapping of Logic Execution for Memristor-Aided logic"?

11. Incorrect sub-figure numbering in Fig. 3 of main text.

Version 1:

Reviewer comments:

Reviewer #1

(Remarks to the Author)

The authors' response indicates that they may be missing the broader perspective and motivation for their work. While I do not doubt the potential value of in-memory computing with memristors or other emerging memory devices and I agree with the first four sentences in Section H of the Supplementary Information, the discussion in that section and the references cited at the end of the paragraph (Refs. 41-47) do not justify the proposed approach.

It is important to highlight a fundamental distinction: Refs. 41–47 report in-memory computing implementations that do not require changing memristors's state each time to compute Boolean logic or dot-product functions. This is a very critical difference and a key point of contention. While I recognize that emerging memory technologies are likely to improve over time, the fundamental challenge remains that write operations in those (nonvolatile, ionic) memory devices are significantly more demanding.

Given this, I strongly suggest that the authors include a convincing discussion of at least one (even if it is niche) application that would benefit from the proposed approach. Alternatively, they should explicitly clarify the advances required in memory (memristor) technology to make their approach competitive with conventional technologies. This could be performed for some commonly used operation, e.g., a full adder circuit.

Awareness of this fundamental issue is crucial as it would provide valuable insights for device and materials scientists (likely including the authors themselves), enabling them to direct their efforts toward the most impactful advancements.

Reviewer #2

(Remarks to the Author)

The authors have addressed my previous concerns with proper discussion and additional results.

I'm satisfied with the revision and recommend potential publication in COMMSSENG.

Reviewer #3

(Remarks to the Author)

The authors have made extensive revisions to the entire manuscript based on my comments and I appreciate their detailed responses and clarifications. The revised manuscript now better highlights the key contributions, distinction with prior art, and benefits of the proposed method. I only two additional comments/suggestions:

- With respect to Table S4, it seems that the area-delay product of the proposed method scales quadratically with the number of inputs in N-bit adder implementations (specifically as $(4N+1) \times (N+4)$), whereas that of Ref 22 (row 5 of Table S4) scales linearly with inputs. From a first look at it, such a quadratic scaling of resources could be quite penalizing for large circuits, especially when such mixed-mode implementations are supposed to be most advantageous for implementing larger/(more complex) circuits. Additionally, it is stated in the main paper that the 4-bit S-Box requires 12 memristors whereas the best alternative implementation requires around 100 transistors. While this is roughly an order better in terms of area, it does not say anything about how the area difference will scale with N for N-bit S-Box nor about the difference in the number of cycles taken (delay). Can the authors comment on this and the overall scaling prospects of their approach for such complex and larger circuits?
- In the VI3 row of Table S1, shouldn't the logic inputs be 2 VI (voltage inputs) and 1 MI (resistance input)?

Version 2:

Reviewer comments:

Reviewer #1

(Remarks to the Author)

Hyperdimensional computing requires very large fan in in-memory computing gates and is not suited for the proposed concept. However, I don't want to delay publication further and I am okay with the current submission.

Reviewer #3

(Remarks to the Author)

Authors have satisfactorily responded to my questions and concerns. I support accepting this manuscript for publication.

Reviewers' comments:

Reviewer #1 (Remarks to the Author):

It seems that the key contributions are threefold. First, it is a “hybrid” memristor Boolean logic that combines previously suggested flavors of memristor logic, most notably including material implication and MAGIC logic. Second, it is a design automation tool for mapping combinational logic circuits to the proposed ones. And third, it is an experimental demonstration of a 4-bit ripple carry adder and 4-bit input/output S-Box, a basic module used in cryptography.

The results are quite heroic and impressive. However, I am missing the big picture and how the proposed work can advance the field of computing.

Authors' response: We appreciate the reviewer's positive remarks on our contributions. To address the concern about the broader impact of our work, we would like to emphasize that in our work, our goal is to present a novel mixed-mode computing paradigm based on memristive devices and supported by a range of carefully orchestrated innovations on device, circuit and architecture level, enabling efficient co-design through design automation with optimality guarantees. On device level, we make full use of rich capabilities of the memristive devices, using them in two different modes. On circuit level, we present and experimentally verify a new computational primitive that extends the earlier MAGIC gate from two to three inputs, thus reducing the number of primitives needed for a given computation. On the architecture level, we are finding the best arrangement of the two above-mentioned modes being mixed on the same physical devices using a novel design automation tool that tightly integrates the new primitives while being extendible to arbitrary further primitives that might be developed in the future. Based on formal methods, it guarantees optimality by providing a mathematical proof that a solution does not exist for given resource restrictions.

Would there be any advantages to implementing adders and S-box modules with the proposed technology compared to the CMOS one, and if so, what are they? For example, on the device level, the demonstrated delay, energy, and energy-delay product for minimum-width NMOS transistors are below 1 ps, 1aJ, and 10^{-31} Js, respectively, for 10nm node technology. The write speed, energy, and their product are many orders of magnitude worse for memristors, even when using optimistic projections. Hence, it seems that the performance of memristor logic, which involves switching resistive states for logic operation, is fundamentally inferior to CMOS logic, and that neglects another serious issue of the limited switching endurance of memristors. So, I would like to see the discussion of specific practical applications that would be enabled by or for which the proposed logic implementation would be superior to the conventional (CMOS) technology.

Authors' response: We appreciate the reviewer's insightful comments on the comparison of memristive devices and CMOS technology. We would like to clarify our perspective as follows:

A primary advantage of memristive devices over CMOS is their ability to combine both logic and memory functions within a single device, which is particularly relevant for logic-in-memory computing schemes. CMOS gates, by contrast, are designed solely for computation and do not have inherent data storage capabilities. As data processing requirements grow, the inefficiencies of moving data back and forth between memory and the CPU in traditional von Neumann architectures become increasingly problematic. Data movement between the memory and processor in CMOS-based systems can consume up to 1000 times more energy than the logic

operations themselves. In fact, this energy limitation is one of the most pressing issues in the computing industry as the energy required to compute grows exponentially while the global energy production grows only linearly, year after year.

In contrast, memristors can perform certain logic operations directly in the memory, thereby eliminating the need for costly data transfers. While we are not system engineers and therefore cannot propose a specific “killer” application, our aim is to demonstrate how computation can be performed on memristive devices, leveraging their unique memory and logic properties. We showcase proof-of-concept applications, such as adders and S-box modules, as examples of computation directly within the memory array. Our design flow allows system engineers to create more complex functions that might be highly promising for logic-in-memory applications. Enabling in-memory computing on a practical level requires a collaborative effort among device engineers and system engineers, and we hope our work will serve as a foundation for future explorations.

As noted in Supplementary Information D, our approach is transferable to a variety of memristive devices, and the performance of our approach is closely tied to the properties of these devices. Although memristors may currently lag CMOS devices in certain aspects, substantial advancements in memristor technology are actively being made. Recent progress includes write speeds reaching 20 ps [1], feature sizes down to $2 \times 2 \text{ nm}^2$ [2], multilevel switching capabilities with over 2048 discrete levels in a single device [3], and stackability of up to 8 layers for 3D integration [4]. These advancements hint at future improvements in speed, energy efficiency, and scalability for memristor-based computing.

Just as CMOS technology evolved from relatively inefficient early implementations to today’s optimized, scaled-down devices, memristor technology is also likely to progress with ongoing research. While memristor-based logic-in memory computing may not yet outperform CMOS for all applications, it is essential to establish foundational designs and approaches for computation-in memory now, which can encourage further development and optimization.

In literature it was shown that hybrid computing concepts offloading some of the processing in the memory, in logic in memory applications [5-6] and in neuromorphic computing applications [7-11], leads to performance improvements in particular by reducing the amount of data to be moved between memory and CPU.

Last but not least, even if it is not the topic of this manuscript, the authors would like to emphasize that another promising application of memristor technology is in neuromorphic computing, where memristor device offers unique capabilities that CMOS transistors cannot match. For instance, a single memristive component can replicate complex synaptic behaviors, such as the Hebbian learning rule and spike-timing-dependent plasticity (STDP) [12-13], as well as the biophysics and temporal dynamics of biological synapses [14], which would otherwise require over 10 transistors in a CMOS circuit [15]. This significantly reduces design complexity and energy consumption. When arranged in a crossbar array, memristors enable efficient, fully connected layers for neural networks, facilitating massively parallel computation. This architecture can perform vector-matrix multiplications—which account for 70-90% of neural network processing—in a single step using Kirchhoff’s laws, thereby executing multiply-and-accumulate operations [16-17]. This results in a 3 to 4 orders of magnitude improvement in computing efficiency over traditional CMOS-based systems.

According to the comment of the reviewer, we have carefully revised the manuscript as follows (on page 21 in Supplementary Information H):

Original text: The energy cost associated with logic computation within the mixed-mode computing paradigm is heavily dependent upon the characteristics of the underlying memristive technology, particularly its switching kinetics. Within a specific device technology, the energy cost is further influenced by the number of both write and readout operations necessary for a given implementation. ...

Revised text: A primary advantage of memristive devices over CMOS is their ability to combine both logic and memory functions within a single device, which is particularly relevant for logic-in-memory computing schemes. As data processing requirements grow, the inefficiencies of moving data back and forth between memory and the CPU in traditional von Neumann architectures become increasingly problematic. Data movement between the memory and processor in CMOS-based systems can consume up to 1000 times more energy than the logic operations themselves. In fact, this energy limitation is one of the most pressing issues in the computing industry as the energy required to compute grows exponentially while the global energy production grows only linearly, year after year. As noted in Supplementary Information D, our approach is transferable to a variety of memristive devices, and the performance of our approach is closely tied to the properties of these devices. Substantial advancements in memristor technology are actively being made. Recent progress includes write speeds reaching 20 ps [37], feature sizes down to $2 \times 2 \text{ nm}^2$ [38], multilevel switching capabilities with over 2048 discrete levels in a single device [39], and stackability of up to 8 layers for 3D integration [40]. These advancements hint at future improvements in speed, energy efficiency, and scalability for memristor-based computing. In literature it was shown that hybrid computing concepts offloading some of the processing in the memory, in logic in memory applications [41-42] and in neuromorphic computing applications [43-47], leads to performance improvements in particular by reducing the amount of data to be moved between memory and CPU.

In the context of the mixed-mode computing paradigm, the energy cost associated with logic computation is heavily influenced by the characteristics of the underlying memristive technology, particularly its switching kinetics. For a specific device technology, this energy cost is further dictated by the number of write and readout operations required for a given implementation. As memristive technology continues to advance, these developments will likely enable even greater efficiency and scalability for mixed-mode and logic-in-memory computing paradigms. ...

My other major concern is that the paper is challenging to read and contains many significant typos (e.g., in panel labeling in Figure 3 and how they are referred to in the text).

Authors' response: We appreciate the reviewer's feedback. In response, we have conducted a thorough review of both the Main Article and the Supplementary Information to identify and correct all typos that we could find, including issues with panel labeling in Figure 3 and their references in the text. The revised version ensures consistency and clarity throughout the manuscript.

Reference list (corresponding to reply to reviewer 1):

[1] Csontos, M., et al. "Picosecond Time-Scale Resistive Switching Monitored in Real-Time." *Advanced Electronic Materials*, 9 (6): 2201104 (2023).

[2] S. Pi, et al., *Nature Nanotechnology* 14, 35 (2019).

[3] M. Rao, et al., *Nature* 615: 823–829 (2023).

- [4] P. Lin, *et al.*, *Nature Electronics* 3, 225 (2020).
- [5] Le Gallo, M., Sebastian, A., Mathis, R. *et al.* Mixed-precision in-memory computing. *Nat Electron* 1, 246–253 (2018).
- [6] M. Le Gallo, A. Sebastian, G. Cherubini, H. Giefers and E. Eleftheriou, "Compressed Sensing With Approximate Message Passing Using In-Memory Computing," in *IEEE Transactions on Electron Devices*, vol. 65, no. 10, pp. 4304-4312, Oct. 2018, doi: 10.1109/TED.2018.2865352.
- [7] Le Gallo, M., Hrynkevych, O., Kersting, B. *et al.* Demonstration of 4-quadrant analog in-memory matrix multiplication in a single modulation. *npj Unconv. Comput.* 1, 11 (2024).
- [8] Ankit, A., El Hajj, I., Chalamalsetti, S. R., Ndu, G., Foltin, M., Williams, R. S., Faraboschi, P., Hwu, W.-m. W., Strachan, J. P., Roy, K., & Milojevic, D. S. (2019). PUMA: A Programmable Ultra-efficient Memristor-based Accelerator for Machine Learning Inference. *Proceedings of the Twenty-Fourth International Conference on Architectural Support for Programming Languages and Operating Systems (ASPLOS '19)*, 715–731. <https://doi.org/10.1145/3297858.3304049>
- [9] Wan, W., Kubendran, R., Schaefer, C. *et al.* A compute-in-memory chip based on resistive random-access memory. *Nature* 608, 504–512 (2022).
- [10] Huang, Y., Ando, T., Sebastian, A. *et al.* Memristor-based hardware accelerators for artificial intelligence. *Nat Rev Electr Eng* 1, 286–299 (2024).
- [11] A. Shafiee *et al.*, "ISAAC: A Convolutional Neural Network Accelerator with In-Situ Analog Arithmetic in Crossbars," *2016 ACM/IEEE 43rd Annual International Symposium on Computer Architecture (ISCA)*, Seoul, Korea (South), 2016, pp. 14-26, doi: 10.1109/ISCA.2016.12.
- [12] W. Huh, D. Lee, C.-H. Lee, "Memristors Based on 2D Materials as an Artificial Synapse for Neuromorphic Electronics," *Advanced Materials* 32: 2002092 (2020).
- [13] Y. Li, K. Su, H. Chen, X. Zou, C. Wang, H. Man, K. Liu, X. Xi, T. Li, "Research Progress of Neural Synapses Based on Memristors," *Electronics* 12: 3298 (2023).
- [14] G.-q. Bi, M.-m. Poo, "Synaptic modifications in cultured hippocampal neurons: dependence on spike timing, synaptic strength, and postsynaptic cell type," *Journal of Neuroscience* 18: 10464-10472 (1998).
- [15] R. Wojtyna, T. Talaśka, "Transresistance CMOS neuron for adaptive neural networks implemented in hardware," *Electronics* 54: (2006).
- [16] D. B. Strukov, K. K. Likharev, "A reconfigurable architecture for hybrid CMOS/nanodevice circuits," *Proceedings of the ACM/SIGDA International Symposium on Field-Programmable Gate Arrays*, pp. 131-140 (2006).
- [17] A. Amirsoleimani, F. Alibart, V. Yon, J. Xu, M. R. Pazhouhandeh, S. Ecoffey, Y. Beilliard, R. Genov, D. Drouin, "In-Memory Vector-Matrix Multiplication in Monolithic Complementary Metal–Oxide–Semiconductor-Memristor Integrated Circuits: Design Choices, Challenges, and Perspectives," *Advanced Intelligent Systems* 2: 2000115 (2020).

Reviewer #2 (Remarks to the Author):

Comment: This manuscript proposed a 3-input logic operation based on the memristors resistive state or operating voltage as input variables through a synergistic design at the device-system level. A mixed-mode in-memory computing method is constructed accordingly. The proposed computational method is experimentally demonstrated using a fabricated self-rectifying memristor array, and its application potential in specific scenarios is evaluated through a ripple carry adder and an S-BOX encryption task. However, the following issues should be addressed before the paper is accepted for publication.

Methodological aspects:

1. The authors need to further explain the differences between the proposed 3-input method and the mentioned two-input method, aside from adding device resistance as an input variable. How does this addition improve the efficiency and accuracy of logical computations? Furthermore, the authors should clearly explain why this 3-input method is more conducive to logic cascading compared to the two-input method.

Authors' response: Thank you for your valuable feedback. We would like to clarify the novel aspects and advantages of our proposed 3-input logic method, which includes the VI^3 and MI^3 operations, and its significance in advancing memristive logic systems through a holistic co-design approach.

- 1. Synergistic Design and Dual Input Variables:** Our 3-input logic method does more than add an extra input—it introduces a synergistic design that integrates two types of 2-input gates (such as the $\bar{p} \cdot q$ and $\bar{p} + q$ gates from the CRS logic family) into a single 3-input operation. This integration simplifies the logic at the circuit level, reducing the design complexity and enabling faster processing per cycle. Furthermore, the 3-input method incorporates both the resistance state and operating voltage as input variables, thus fully leveraging the unique dual physical properties of memristors: the resistance stored in each cell and the applied voltage at the electrodes. This mixed-mode approach to computation allows us to achieve more efficient and accurate logic operations without requiring additional readout logic.
- 2. Efficiency in Logic Cascading:** The 3-input method is inherently conducive to logic cascading, offering a key advantage over the two-input style, which requires the memristive device to be initialized before each operation. In a two-input configuration, cascading logic functions would necessitate an additional re-initialization step, as the result of one operation cannot directly serve as the input for the next without resetting the device. Our 3-input approach, however, allows the output from one operation to be immediately used as the input for subsequent operations, eliminating the need for re-initialization. This direct integration enhances the overall throughput and enables a more streamlined and efficient design, reducing the number of sequential stages required and facilitating a smooth flow of operations.
- 3. Enhanced Automation Tool Support:** The 3-input method is inherently advantageous for logic cascading, as it reduces the number of sequential stages needed, which directly translates to a more streamlined circuit design and enhanced throughput. Additionally, this approach simplifies the development of automation tools at the software level, enabling more complex logic functions to be implemented more easily and quickly. The 3-input logic design reduces the need for separate stages or additional complexity in tool design, making it an efficient and practical choice for memristive systems.

4. **Holistic Co-Design Approach:** Our study emphasizes a co-design approach, integrating device, circuit, and system levels for optimal performance. Addressing any single level in isolation would overlook the interdependent advancements that could significantly enhance the system's capabilities. In our approach, the 3-input logic design supports the automation tool design, while the automation tool design, in turn, facilitates efficient implementation of the 3-input logic. This coordinated approach marks a key advancement that we believe can inspire further innovation within the field.

We hope these clarifications highlight the unique contributions of our work. We have included this detailed explanation in the manuscript to make the conceptual foundation, technical benefits, and broader implications of our design clear to readers across research disciplines. Thank you for helping us improve the clarity and impact of our manuscript.

We believe this response and clarification convey the distinct contributions of our work, which we incorporate into the manuscript as follows for clearer understanding (on page 5 in Supplementary Information A):

Original text: “In this work, we propose a 3-input VI^3 logic operation, representing an extended iteration of CRS logic (without readout). Unlike conventional CRS logic design, this approach expands the scope of input variables from 2 to 3 and establishes a complete gate set through integration with MI^3 logic operation, thereby eliminating the need for a readout operation. This elimination of the readout operation during logic cascading is a significant advantage of our work.”

Revised text: “In this work, we propose a 3-input VI^3 logic operation, representing an extended iteration of CRS logic (without readout). Unlike conventional CRS logic design, this approach expands the scope of input variables from 2 to 3 and establishes a complete gate set through integration with MI^3 logic operation, thereby eliminating the need for a readout operation. This elimination of the readout operation during logic cascading is a significant advantage of our work, which will be detailed further in Supplementary Information C. Besides that, the 3-input logic operations proposed in this work offer several distinct advantages: Firstly, they embody a synergistic design by integrating two types of 2-input gates, such as the $\bar{p} \cdot q$ and $\bar{p} + q$ gates from the CRS logic family, into a single 3-input operation. This integration reduces circuit-level design complexity and enables faster processing per cycle. Additionally, the 3-input configuration fully exploits the dual physical properties of memristors—stored resistance and applied electrode voltage—by using both resistance states and operating voltages as input variables. This approach achieves efficient and accurate logic operations without requiring additional readout logic, significantly improving overall performance. Secondly, the 3-input design provides substantial efficiency gains in logic cascading. Unlike traditional 2-input gates that require memristive devices to be re-initialized before each operation, the 3-input design allows the output from one operation to be immediately reused as input in subsequent operations, bypassing re-initialization steps. This direct integration enhances throughput, reduces the number of sequential stages, and supports a streamlined and efficient design flow, ensuring smooth operation between MI and VI kernels. Furthermore, the 3-input configuration simplifies the implementation of complex logic functions within automation tools by reducing the need for additional stages and design complexity. This makes it a practical solution for managing memristive systems in crossbar arrays, improving both automation tool development and overall circuit design. The last but not the least, our work adopts a holistic co-design methodology that integrates device, circuit, and system-level considerations to maximize performance. By addressing these levels in a

coordinated manner, the approach leverages interdependencies that enhance system capabilities. The 3-input logic design optimizes automation tool development, which in turn facilitates the effective implementation of the design within crossbar architectures, achieving a high degree of efficiency, flexibility, and functionality.”

2. The authors should provide a clearer explanation of the working principle of the mapping synthesis tool. The description of equation (3) is not easily understood. It might be helpful to include pseudo-code or a program flowchart. If possible, the authors should also explain how the logic synthesis tool is used to synthesize specific application examples in the subsequent application section.

Authors’ response: Thank you for the insightful suggestion! In response to the suggestion of reviewer, to further illustrate the automation flow tool’s functionality, we have included a flowchart in the Supplementary Information F. This flowchart visually demonstrates the stages involved in synthesizing specific logic circuits, such as adders, to provide readers with an intuitive understanding of the process. We have inserted the flow chart and the illustration of Eq. 1 in Fig. S8 in Supplementary information F and carefully revised the manuscript as follows:

Newly added descriptions (on page 17 in Supplementary Information F): The flowchart of the M³S automation tool is demonstrated in Fig. 8, using an example of the VI kernel and the construction of Eq. 1. As shown in the flowchart, the process begins with the construction of a Boolean formula in conjunctive normal form format, which is input into a Boolean satisfiability solver. The solver determines whether a solution exists and, if successful, the automation tool interprets the solution and implements it on the memristive cells within the crossbar.

Fig. S8 (a) Flowchart illustrating the working principle of the M³S automation tool for synthesizing and mapping logic circuits on memristive crossbars. The constructed Boolean formula, expressed in conjunctive normal form (CNF), serves as input to a Boolean satisfiability (SAT) solver to determine a solution. (b) Illustration of the V-mode operation in Eq. 1, providing an example of literal functions for n = 3 inputs. The table outlines the function table entries for each variable $l_{j,q}$ in the formula, demonstrating how the SAT solver assigns one variable for the top electrode (TE) and one for the bottom electrode (BE) to control which of the (2n+2) literal functions is applied during each cycle. For each crossbar cell M_{ab} in cycle c , the Boolean satisfiability solver ensures that exactly one variable $g_{a,b,j,c}^{TE}$ is set to 1 for the top electrode and exactly one variable $g_{a,b,k,c}^{BE}$ is set to 1 for the bottom electrode, while all other variables are set to 0. This precise selection determines which of the (2n+2) literal functions is applied during that cycle, enabling accurate control of the memristive logic operation.

3. In terms of application examples, the authors need to present results in a clearer way. For example, in the application of the ripple carry adder, it would be better to indicate which data are being added together and what result is obtained. The current plotting figures are not convenient for readers to understand.

Authors' response: Thank you for your feedback on clarifying the presentation of application examples. We have provided a more detailed breakdown of the ripple carry adder (CRA) implementation to improve readability and understanding. Specifically, we have made the following enhancements:

In the manuscript, we now explain the operation of the carry ripple adder (CRA) in a step-by-step manner, using an example where the inputs are $a = 0011$ and $b = 0101$, corresponding to individual bits $a_3 = 0, a_2 = 0, a_1 = 1, a_0 = 1$ and $b_3 = 0, b_2 = 1, b_1 = 0, b_0 = 1$ (with $a_0 = 1$ and $b_0 = 1$ as an example). This example demonstrates how the addition proceeds in a bit-by-bit sequence, highlighting the processing flow and the final result. This clarification is intended to make the application example easier to follow and understand for readers.

We would like to draw reviewers' attention that we have extensive explanation in the Supplementary Information E to explain the high-parallel computing approach used for the adder application in the memristive crossbar. This section outlines various parallel strategies we considered, with a specific focus on the carry ripple adder implementation in Figure 3 of the main text. In particular, Figure S6 in the Supplementary Information provides an illustration of the implementation flow for a N-bit full adder, including the implementation flow in each cell in each cycle.

a 1-bit full adder based on standard logic gates

b Control sequence of 1-bit full adder

c $a = x_{20}, b = x_{30}, c = x_{10}$

Cycle	M11	M12	M13	M14	M15
BL_1 (TE)	x_{10}	x_{10}	x_{10}	x_{10}	x_{10}
1	x	x	x	x	x
WL_1 (BE)	\bar{x}_{10}	\bar{x}_{10}	\bar{x}_{10}	\bar{x}_{10}	\bar{x}_{10}
BL_1 (TE)	x_{30}	\bar{x}_{30}	\bar{x}_{30}		x_{30}
2	c_0	c_0	c_0	c_0	c_0
WL_1 (BE)	\bar{x}_{20}	\bar{x}_{20}	\bar{x}_{20}		\bar{x}_{20}
BL_1 (TE)		1	0		
3	c_1	v_{40}	v_{70}	c_0	c_1
WL_1 (BE)		\bar{x}_{30}	\bar{x}_{30}		
BL_1 (TE)	GND		V_{in}	V_{in}	
4	c_1	v_{50}	v_{80}	c_0	c_1
WL_1 (BE)					
BL_1 (TE)	V_{in}	GND		V_{in}	
5	s'_0	v_{50}	v_{80}	c_0	c_1
WL_1 (BE)					
BL_1 (TE)		V_r			
Verification	s'_0	s_0	v_{80}	c_0	c_1
WL_1 (BE)		GND			

d Physical implementation in crossbar

Fig. S6. Implementation of 1-bit full adder using (a) standard logic gates (NOR gate) in comparison to the design using (b) mixed-mode computing. (c) Step-by-step control cycles for applying logic

inputs through BLs and WLs in the crossbar configuration. (d) Physical implementation in memristor-based crossbar for each cycle.

According to reviewer's comments, we further added Figure S6 in the Supplementary Information. This figure includes a control sequence for a 1-bit CRA using a BFO memristive crossbar of size 1x5, showing how a 4x5 passive crossbar array realizes the adder function. To further improve clarity, we added a detailed illustration in Supplementary Information E, where we explain the 1-bit full adder's operation cycle-by-cycle. Specifically: Figure S6 illustrates the automation flow for the 1-bit CRA, with each column representing a cell (M11–M14) needed for computation, while M15 stores the output carry bit c1. The five columns represent the five cycles required for completing the 1-bit CRA, broken down into three cycles for VI^3 operations and two for MI^3 operations. The VI^3 operations are executed across all cells simultaneously, enabling efficient processing. For each cycle, we specify the WL (Word Line) and BL (Bit Line) inputs used. Notably, while the 1-bit CRA requires five cycles, an N-bit CRA requires only N+1 cycles. This allows for efficient, high-speed processing, as each additional bit requires only one additional cycle, illustrating the scalability and efficiency of our design.

According to the comment of the reviewer, we have inserted Fig. S6 into the SI newly and carefully revised the manuscript as follows (on page 14 in Supplementary Information E):

Original text: "For example, as demonstrated in Fig. S7, the 4-CRA can be realized in a 4x5 passive crossbar array with 1R configuration, where no readout is allowed in logic cascading. The computation starts with a VI^3 by programming all cells into a known state c0, which in principle allows computation to be started with arbitrary logic inputs."

Revised text: "For example, as demonstrated in Fig. S6, in the case of the 1-bit carry-ripple adder, each column representing a cell (M11–M14) needed for computation, while M15 stores the output carry bit c1. The five columns represent the five cycles required for completing the 1-bit CRA, broken down into three cycles for VI^3 operations and two for MI^3 operations. The VI^3 operations are executed across all cells simultaneously, enabling efficient processing. For each cycle, we specify the WL (Word Line) and BL (Bit Line) inputs used. For implementing 1-bit carry-ripple adder, our mixed-mode approach requires 5 cells and 5 cycles, while a purely in-memory computing approach without reading would require 10 cells and 13 cycles (as referenced in Ref. [22] in Tab. S4 in Supplementary Information I). However, the efficiency gains of the mixed-mode paradigm become more apparent as the bit number increases. For instance, in Fig. S7, the 4-CRA can be realized in a 4x5 passive crossbar array with 1R configuration, where no readout is allowed in logic cascading. Implementing an 8-bit full adder in our mixed-mode approach requires only 33 cells and 12 cycles, compared to a purely in-memory computing approach, which would require 87 cells and 97 cycles (also as referenced in Ref. [22] in Tab. S4 in Supplementary Information I). This comparison highlights the efficiency of mixed-mode computing in terms of cell and cycle optimization, particularly for multi-bit and complex operations."

4. Can the authors provide a quantitative performance improvement for the application examples? For instance, compared to CMOS and existing memristor logic computation methods, how much improvement in computational energy efficiency and speed can be achieved at different bit widths?

Authors' response: We appreciate the reviewer's insightful comments regarding quantitative performance comparisons between memristive devices and CMOS technology. The primary advantage of memristive devices lies in their ability to integrate both logic and memory functions

within a single device, which is particularly advantageous for logic-in-memory computing schemes. In contrast, CMOS gates are designed solely for computation and lack inherent data storage capabilities. This separation in traditional von Neumann architectures results in inefficiencies, as data transfers between memory and processors can consume up to 1000 times more energy than the logic operations themselves. Memristors address this limitation by performing logic operations directly within memory, significantly reducing the energy cost associated with data movement.

Our work demonstrates proof-of-concept applications, such as adders and S-box modules, showcasing computation within memory arrays. While our focus is on enabling foundational designs rather than proposing a specific “killer” application, these examples highlight the potential of memristor-based logic in-memory computing. These designs can be further expanded by system engineers to create more complex functions for practical applications. The realization of in-memory computing at scale requires close collaboration between device engineers and system engineers, and we hope our study serves as a foundation for such developments.

As discussed in Supplementary Information D, the performance of our approach is directly linked to the properties of the underlying memristive devices. Recent advancements in memristor technology are promising, with reported write speeds reaching 20 ps [1], feature sizes as small as $2 \times 2 \text{ nm}^2$ [2], multilevel switching with over 2048 discrete levels [3], and stackability of up to 8 layers for 3D integration [4]. These developments suggest that significant improvements in speed, energy efficiency, and scalability for memristor-based computing are achievable as the technology matures.

While memristor technology may not yet outperform CMOS in all applications, it is important to consider that CMOS itself evolved from relatively inefficient early implementations to today’s optimized systems. Similarly, as memristor technology progresses, it is expected to become increasingly competitive. Establishing foundational designs for logic in-memory computing now is critical to fostering further research and optimization.

In addition to logic in-memory computing, memristors also offer unique advantages in neuromorphic computing, an area where CMOS technology cannot match their capabilities. For instance, a single memristive device can replicate complex synaptic behaviors such as the Hebbian learning rule and spike-timing-dependent plasticity (STDP) [5,6], as well as the temporal dynamics of biological synapses [7]. Implementing these features in CMOS circuits typically requires more than 10 transistors [8], leading to significantly higher design complexity and energy consumption. When arranged in crossbar arrays, memristors enable efficient, fully connected layers for neural networks, performing vector-matrix multiplications in a single step using Kirchhoff’s laws [9,10]. This approach allows for multiply-and-accumulate (MAC) operations with 3 to 4 orders of magnitude improvement in computational efficiency compared to CMOS-based systems.

Lastly, existing literature demonstrates that hybrid computing concepts, which offload some processing into memory for both logic in-memory computing [11,12] and neuromorphic applications [13–17], yield substantial performance improvements. These improvements are primarily achieved by minimizing data movement between memory and the CPU. Although quantitative benchmarks (e.g., energy efficiency and speed at varying bit widths) for the proposed approach are beyond the scope of this study, our work establishes a strong foundation for such future analyses as memristor technology continues to advance.

According to the comment of the reviewer, we have carefully revised the manuscript as follows (on page 21 in Supplementary Information H):

Original text: The energy cost associated with logic computation within the mixed-mode computing paradigm is heavily dependent upon the characteristics of the underlying memristive technology, particularly its switching kinetics. Within a specific device technology, the energy cost is further influenced by the number of both write and readout operations necessary for a given implementation. ...

Revised text: A primary advantage of memristive devices over CMOS is their ability to combine both logic and memory functions within a single device, which is particularly relevant for logic-in-memory computing schemes. As data processing requirements grow, the inefficiencies of moving data back and forth between memory and the CPU in traditional von Neumann architectures become increasingly problematic. Data movement between the memory and processor in CMOS-based systems can consume up to 1000 times more energy than the logic operations themselves. In fact, this energy limitation is one of the most pressing issues in the computing industry as the energy required to compute grows exponentially while the global energy production grows only linearly, year after year. As noted in Supplementary Information D, our approach is transferable to a variety of memristive devices, and the performance of our approach is closely tied to the properties of these devices. Substantial advancements in memristor technology are actively being made. Recent progress includes write speeds reaching 20 ps [37], feature sizes down to $2 \times 2 \text{ nm}^2$ [38], multilevel switching capabilities with over 2048 discrete levels in a single device [39], and stackability of up to 8 layers for 3D integration [40]. These advancements hint at future improvements in speed, energy efficiency, and scalability for memristor-based computing. In literature it was shown that hybrid computing concepts offloading some of the processing in the memory, in logic in memory applications [41-42] and in neuromorphic computing applications [43-47], leads to performance improvements in particular by reducing the amount of data to be moved between memory and CPU.

In the context of the mixed-mode computing paradigm, the energy cost associated with logic computation is heavily influenced by the characteristics of the underlying memristive technology, particularly its switching kinetics. For a specific device technology, this energy cost is further dictated by the number of write and readout operations required for a given implementation. As memristive technology continues to advance, these developments will likely enable even greater efficiency and scalability for mixed-mode and logic-in-memory computing paradigms. ...

Experimental aspects:

5. The device data needs to be supplemented. Since the authors rely on the device's resistive state switching to perform logic operations, the characteristics of the device's resistive state retention, endurance, variation, switching ratio, etc., will all impact the performance of logical computations. Therefore, the authors should supplement this data.

Authors' response: We appreciate the reviewer's comment regarding the need for additional device data to supplement the work. In response, we emphasize that the IV characteristics, switching dynamics, and underlying switching mechanisms of the BFO memristor, as well as its variants—Ti-doped BFO memristor (BiFeTiO₃, BFTO) and bilayer BFO memristor (BiFeTiO₃/BiFeO₃, BiBFO)—are detailed in Supplementary Information B. These data provide a

comprehensive comparison of the devices and establish their suitability for performing logic operations. Such detailed characterization is fundamental to the design of in-memory computing systems based on memristor technology.

Herewith we would like to emphasize that the goal of our work adopts a holistic co-design methodology for achieving high-performance in-memory computing systems. This approach integrates innovations across device, circuit, architecture, and design automation levels, ensuring that all aspects operate synergistically to enhance performance, efficiency, and scalability. This makes our work distinct from most state-of-the-art approaches that typically focus on one, maximum two levels. We believe this co-design strategy provides a robust foundation for future high-performance in-memory computing systems.

While our work highlights the capabilities of BFO memristors as an exemplary proof-of-principle, it is essential to note that the methodology is transferable to a wide range of memristive devices and technologies, as discussed in Supplementary Information D. Therefore, while the device-specific characteristics such as retention, endurance, switching ratio, and variations inevitably impact the performance of logical computations, the focus of our current work is *not* on the specific device-level performance but rather on demonstrating the overarching co-design framework.

To address the reviewer's request, we have added descriptions in Supplementary Information B to explicitly reference earlier publications where these performance metrics (e.g., retention, endurance, switching ratio, and variations) of the BFO memristor and its variants are comprehensively analyzed [18-22]. Furthermore, we consider an in-depth evaluation of the impact of individual device characteristics on logical computations as part of future work to build upon this foundational study.

→ *Newly added text* (on page 8, in Supplementary Information B): The performance metrics, including retention, endurance, switching ratio, and variations of the BFO memristor and its variants, have been comprehensively analyzed and published in our previous works [16-20].

6. It would be beneficial for the authors to analyze the impact of non-ideal effects of the device on the proposed logic computation method. Which factor has the greatest impact? What effect does it have on the logic computation results? Compared to existing computation methods, does the proposed 3-input method offer better robustness?

Authors' response: The non-ideal effects of the device significantly influence the performance of the proposed 3-input logic computation method, particularly in MI³ and VI³ operations. As discussed in the main article and Supplementary Information A, the greatest impact arises from the switching dynamics and stochastic variability inherent in nanoscale memory devices. Switching dynamics dictate the types of logic gates that can be implemented, with some gates requiring additional processes such as RESET or SET operations, leading to prolonged operation sequences and reduced efficiency. Stochastic variability further exacerbates the issue by introducing high error rates—up to 70% in experimental studies—particularly in cascaded operations. These effects collectively reduce the scalability and accuracy of the computation method. Additional challenges include input drift and partial switching in stateful designs, which can compromise the robustness of MI³ operations.

Compared to existing computation methods, the proposed mixed-mode 3-input logic approach demonstrates superior robustness by combining the strengths of MI and VI kernels while mitigating their individual limitations. MI kernels, though capable of universal logic, often suffer

from long processing cycles and high error rates due to device variability. On the other hand, VI kernels, which rely on voltage inputs rather than stateful logic, are not inherently universal and require additional readout circuitry to implement arbitrary functions, leading to increased latency and power consumption. The mixed-mode framework leverages the complementary advantages of these kernels to achieve universality without additional readout operations. This integration results in shorter processing cycles and enhanced resilience, with functionality shifted to the more reliable VI³ operations.

The use of BFO memristors in this work further enhances the robustness of the MI³ kernel. These devices exhibit unique self-rectifying analog bipolar switching behavior, which addresses common non-idealities such as input drift and partial switching in stateful logic designs. By optimizing the programming bias ($V_{in} = 5.3\text{ V}$), the method ensures accurate transitions in MI³ operations while enabling reliable cascading (to be detailed in the reply to Comment 7). This feature improves computing accuracy, particularly in practical implementations of cascaded MI³ operations.

While the current work highlights these benefits, a detailed quantitative analysis of non-ideal effects on the MI³ operations using BFO memristors is beyond its scope. This aspect will be explored further as part of future research in a separate manuscript. Overall, the proposed method demonstrates clear advantages in robustness and efficiency compared to existing approaches, particularly in scenarios requiring deeply cascaded operations.

According to the comment of the reviewer, we have carefully revised the manuscript as follows (on page 11 in Supplementary Information C):

Original text: “Next, using BFO-based MI³ operation as an example (with experimental data shown in Fig. S4), we discuss two major issues in logic designs within the MI kernel, i.e. “Input drift” and “Partial switching” issues. Both issues are inevitable in stateful logic operations. Notably, with much more significant device variability issue in abrupt switching devices compared to analog ones, it further reduces the programming window and leads to an even higher error rate in logic processing.”

Revised text: “The non-ideal effects of the device significantly influence the performance of the proposed 3-input logic computation method, particularly in MI³ and VI³ operations. As discussed in the main article and Supplementary Information A, the greatest impact arises from the switching dynamics and stochastic variability inherent in nanoscale memory devices. Switching dynamics dictate the types of logic gates that can be implemented, with some gates requiring additional processes such as RESET or SET operations, leading to prolonged operation sequences and reduced efficiency. Stochastic variability further exacerbates the issue by introducing high error rates—up to 70% in experimental studies—particularly in cascaded operations. These effects collectively reduce the scalability and accuracy of the computation method. Next, using BFO-based MI³ operation as an example (with experimental data shown in Fig. S4), we discuss two major issues in logic designs within the MI kernel, i.e. “Input drift” and “Partial switching” issues. Both issues are inevitable in stateful logic operations. Notably, with much more significant device variability issue in abrupt switching devices compared to analog ones, it further reduces the programming window and leads to an even higher error rate in logic processing.”

7. The authors used SRM devices for experimental verification. Does the use of SRM devices, compared to bipolar devices or 1T1R devices, contribute to enhancing the performance of the 3-input logic computation method? In what aspects specifically?

Authors' response: Thank you for this insightful question. Yes, in this work, we adopt a technology-aware co-design strategy, i.e. leveraging the unique analog self-rectifying switching dynamics of BFO memristors to optimize MI³ operations and enhance the overall computing performance. It is important to note that each memristive technology offers distinct advantages, and a technology-aware design process is crucial for achieving optimized performance tailored to the specific technology used.

As discussed in Supplementary Information A, MI-based logic designs, including IMPLY, MAGIC, FELIX, and the proposed MI³ operation, share general drawbacks such as partial switching and drift issues due to the voltage divider effect. To mitigate these, we exploit the analog self-rectifying switching behavior of BFO memristors, which significantly enhances computational accuracy.

In our prior work (Ref. [23]), by leveraging the unique analog self-rectifying switching dynamics of BFO memristors, we demonstrated that the realization of stateful logic gates, such as MAGIC NOR, is strongly influenced by the underlying technology's switching behavior. For instance, Ref. [24] highlights that implementing MAGIC NOR with digital bipolar memristors typically requires a SET voltage greater in magnitude than the RESET voltage. In contrast, BFO memristors, with equal SET and RESET voltage magnitudes, enable the successful implementation of MAGIC NOR due to their self-rectifying behavior. This underscores the importance of technology-specific analysis and optimization in effectively adapting different memristive technologies for logic design.

Based on that, in this work, we exploit the unique analog self-rectifying switching behavior of BFO memristors to achieve optimized performance of MI³. Unlike digital memristors, which switch abruptly when a threshold bias or current is exceeded, BFO memristors exhibit gradual current changes due to their analog nature. This characteristic makes them less sensitive to device variations and capable of processing correct outputs in MI kernel operations. However, the partial switching caused by the voltage divider effect remains a challenge in analog memristor-based logic designs.

To address this, we analyzed the switching kinetics of BFO memristors, which favor easier transitions from LRS to HRS compared to HRS to LRS. Based on this analysis, we structured the MI³ operation to apply a positive bias to the top electrodes (TEs) of two parallel-connected memristors while grounding the third memristor's TE. This configuration is beneficial for us to exploit the cascading process as one re-initialization process to output cell with logic '1', which minimizes unwanted drift in input cells and enhances the stability of MI³ kernel logic operations. For example, in this work, we optimized the output to reliably produce a precise '0' state by setting the bias voltage (V_{in}) to 5.3V. This voltage ensures stable '0' outputs across various input combinations, reducing errors in cascaded operations. Then we exploit the cascading process to re-initialize intermediate output '1'.

In addition, to further enhance cascaded logic operations, our M³S automation tool is designed to prevent the output cell of one MI³ cycle from being reused as an output cell in subsequent cycles, thereby minimizing error rates due to drift accumulation. This mixed-mode design reassigns the output cell as an input cell in subsequent operations, maintaining higher operational accuracy.

According to the comment of the reviewer, we have carefully revised the manuscript as follows (on page 12 in Supplementary Information C):

Original text: “In mixed-mode computing, we not only exploit variation-resilient VI^3 for carrying out partial processing but also design the advanced MI^3 to enable robust cascading with multiple MI^3 or VI^3 operations. To achieve this, we employ $V_{in} = 5.3$ V, completely avoiding the “input drift”—the change of memristance states in input cells (see experimental results in Fig. S3, while ensuring correct transitions from LRS to HRS in the MI^3 operation with input combinations ‘101’, ‘110’, and ‘111’ (HRS of M1 after MI^3 : 111.4 M Ω compared to initialized HRS of M1 at 402.4 M Ω) as demonstrated in Fig. S3. As expected, with $V_{in} = 5.3$ V, the MI^3 operation with input combination ‘100’ results in a compromised LRS in M1 after MI^3 (LRS of M1 after MI^3 : 27.2 M Ω compared to initialized LRS of M1 at 1.1 M Ω). Additionally, in the M^3S automation tool, we insert conditions to prioritize using the output cell as an input cell in cascaded MI^3 (or VI^3) operations. Hence, the compromised LRS in M1 can be used as an input cell, where the compromised LRS is refreshed back to logic ‘1’ by the positive bias V_{in} to the TE of M1, while ensuring correct output results, thereby significantly reducing the error rate and enhancing the robustness of MI^3 operations during deep cascading. “

Revised text: “Both aforementioned “input drift” and “partial switching” issues are inherent in the stateful logic process (cannot be alleviated). Under mixed-mode computing paradigm, our aim is to mitigate these issues through a co-design strategy that enhances computing accuracy, particularly by leveraging the unique self-rectifying and analog switching characteristics of BFO memristors. This emphasizes the critical importance of technology-specific optimization in adapting memristor devices for logic designs.

Firstly, to ensure stable and accurate outputs, we carefully analyzed the switching kinetics of BFO memristors. These kinetics favor easier transitions from LRS to HRS compared to HRS to LRS. Based on this, we structured the MI^3 operation to apply positive bias to the TEs of two parallel-connected memristors while grounding the TE of the third memristor. This configuration allows us to exploit cascading for “re-initializing” the logic ‘1’, minimizes the influence of partial switching in the input cells, and enhances stability within the cascading MI kernel. Consequently, we designed the MI^3 operation to produce a more reliable ‘0’ output by optimizing the bias voltage (V_{in}) to 5.3 V. For instance, as demonstrated in experimental results in Fig. S3, the MI^3 operation demonstrates correct transitions from LRS to HRS with input combinations ‘101’, ‘110’, and ‘111’ (HRS of M1 after MI^3 : 111.4 M Ω compared to initialized HRS of M1 at 402.4 M Ω) as shown in Fig. S3. As expected, with $V_{in} = 5.3$ V, the MI^3 operation with input combination ‘100’ results in a compromised LRS in M1 after MI^3 (LRS of M1 after MI^3 : 27.2 M Ω compared to initialized LRS of M1 at 1.1 M Ω). Thus this bias $V_{in} = 5.3$ V ensures stable ‘0’ outputs across various input combinations, reducing errors in cascaded operations (see experimental results in Fig. S3). As next step, in the definition of M^3S automation tool is explicitly designed to allow the output cell of MI^3 operation can only be reassigned as an output cell in subsequent MI^3 operations if it has been used at least once as an input cell in an MI^3 or VI^3 operation. This intentional reassignment avoids scenarios where the output ‘1’ from a prior MI^3 gate is recomputed into another ‘1’ in the next MI^3 cycle, ensures that a more precisely defined input or output value is applied in logic cascading, thereby minimizing error propagation commonly observed in traditional stateful approaches.”

8. The authors mentioned that this method can be transferred to different devices such as STT-MRAM. Does the performance of the 3-input logic method vary on different devices? Which one do the authors most recommend as the computing medium for this method, and why?

Authors' response: Yes, the performance of the 3-input logic method can vary depending on the underlying device technology. This variation is influenced by factors such as the switching dynamics of the technology, which determine the types of logic gates that can be realized, and device variability, which affects logic accuracy, especially for resistance-controlled logic design.

To adopt the proposed mixed-mode computing architecture, the chosen technology must satisfy two essential criteria: (1) it should enable resistance-controlled logic to support mixed-mode computing, and (2) it should exhibit low device variability to achieve high logic accuracy. Additionally, good endurance is also an important consideration for practical applications.

While the selection of the optimal device largely depends on the target application, in general, technologies that offer higher density, lower energy consumption, and faster operation would be most desirable. At this point, we cannot give a specific recommendation as this requires specific device knowledge. This would be an interesting aspect for future research. This would be an interesting aspect for future research.

According to the comment of the reviewer, we have carefully revised the manuscript as follows (on page 14 in Supplementary Information D):

Original text: “Table S2 demonstrates that the same 3-input V_I^3 logic function can be implemented using various series of BFO memristors, as well as more general types of redox-based random access memory (ReRAM), phase change memory (PCM), and spin transfer torque magneto-resistive RAM (STT-MRAM). This implementation involves customizing the definitions of v_{x2} and v_{x3} to initiate switching based on the specific characteristics of each underlying technology.”

Revised text: “It is worthy to mention that the performance of the 3-input logic method can vary depending on the underlying device technology. This variation is influenced by factors such as the switching dynamics of the technology, which determine the types of logic gates that can be realized, and device variability, which affects logic accuracy, especially for resistance-controlled logic design. Table S2 demonstrates that the same 3-input V_I^3 logic function can be implemented using various series of BFO memristors, as well as more general types of redox-based random access memory (ReRAM), phase change memory (PCM), and spin transfer torque magneto-resistive RAM (STT-MRAM). This implementation involves customizing the definitions of v_{x2} and v_{x3} to initiate switching based on the specific characteristics of each underlying technology. To adopt the proposed mixed-mode computing architecture, the chosen technology must satisfy two essential criteria: (1) it should enable resistance-controlled logic to support mixed-mode computing, and (2) it should exhibit low device variability to achieve high logic accuracy. Additionally, good endurance is also an important consideration for practical applications. While the selection of the optimal device largely depends on the target application, in general, technologies that offer higher density, lower energy consumption, and faster operation would be most desirable. “

Grammar and writing aspects:

9. The authors should reduce the use of abbreviations. The extensive use of abbreviations and the confusion of various concepts make the paper difficult to understand and obscure the presentation of the paper's highlights.

Authors' response: According to the suggestion from the reviewer, we have largely reduced the use of abbreviations, e.g. BE, TE, N-CRA, SAT, CNF, BFO, BFTO, and BiBFO.

10. Can equation (3) be broken down into a set of equations to explain the working principle of the synthesis tool? The current equation (3) is difficult to grasp quickly.

Authors' response: It is difficult to express Equation 3 by simpler equations, because it integrates all information related to VI^3 on M_{78} and leaving out parts of it would skip essential dependencies. It is true that Equation 3 is internally transferred into conjunctive normal form, that is, a number of disjunctive clauses (OR-expressions of variables and negated variables), iterated for all possible values of iterators i, j, k and q under the conjunction sign. Showing them would create space problems, as their number would be excessive, and they would be not illustrative as they would lack internal logic contained in the current Equation 3 (implications and equivalencies would be translated into uniformly-looking clauses. Note that the solving is not done by processing parts of equation (or clauses) one-by-one, but by assigning variables until all clauses are satisfied or unsatisfiability is proven. We do appreciate the wish to make the Boolean satisfiability model understandable, and for this reason we already decided to not show the complete formula but one equation of it for one operation on one specific cell; the complete formula has more similarly-looking equations for other operations.

11. Page 5, Line 18 contains a spelling error.

Authors' response: We thank the reviewer for pointing out this. We have revised the text accordingly (from "celT" to "cell").

Reference list (reply to reviewer 2)

- [1] Csontos, M., et al. "Picosecond Time-Scale Resistive Switching Monitored in Real-Time." *Advanced Electronic Materials*, 9 (6): 2201104 (2023).
- [2] S. Pi, et al., *Nature Nanotechnology* 14, 35 (2019).
- [3] M. Rao, et al., *Nature* 615: 823–829 (2023).
- [4] P. Lin, et al., *Nature Electronics* 3, 225 (2020).
- [5] W. Huh, D. Lee, C.-H. Lee, "Memristors Based on 2D Materials as an Artificial Synapse for Neuromorphic Electronics," *Advanced Materials* 32: 2002092 (2020).
- [6] Y. Li, K. Su, H. Chen, X. Zou, C. Wang, H. Man, K. Liu, X. Xi, T. Li, "Research Progress of Neural Synapses Based on Memristors," *Electronics* 12: 3298 (2023).
- [7] G.-q. Bi, M.-m. Poo, "Synaptic modifications in cultured hippocampal neurons: dependence on spike timing, synaptic strength, and postsynaptic cell type," *Journal of Neuroscience* 18: 10464-10472 (1998).
- [8] R. Wojtyna, T. Talaśka, "Transresistance CMOS neuron for adaptive neural networks implemented in hardware," *Electronics* 54: (2006).
- [9] D. B. Strukov, K. K. Likharev, "A reconfigurable architecture for hybrid CMOS/nanodevice circuits," *Proceedings of the ACM/SIGDA International Symposium on Field-Programmable Gate Arrays*, pp. 131-140 (2006).
- [10] A. Amirsoleimani, F. Alibart, V. Yon, J. Xu, M. R. Pazhouhandeh, S. Ecoffey, Y. Beilliard, R. Genov, D. Drouin, "In-Memory Vector-Matrix Multiplication in Monolithic Complementary Metal–

Oxide–Semiconductor-Memristor Integrated Circuits: Design Choices, Challenges, and Perspectives," *Advanced Intelligent Systems* 2: 2000115 (2020).

[11] Le Gallo, M., Sebastian, A., Mathis, R. *et al.* Mixed-precision in-memory computing. *Nat Electron* **1**, 246–253 (2018).

[12] M. Le Gallo, A. Sebastian, G. Cherubini, H. Giefers and E. Eleftheriou, "Compressed Sensing With Approximate Message Passing Using In-Memory Computing," in *IEEE Transactions on Electron Devices*, vol. 65, no. 10, pp. 4304-4312, Oct. 2018, doi: 10.1109/TED.2018.2865352.

[13] Le Gallo, M., Hrynkevych, O., Kersting, B. *et al.* Demonstration of 4-quadrant analog in-memory matrix multiplication in a single modulation. *npj Unconv. Comput.* **1**, 11 (2024).

[14] Ankit, A., El Hajj, I., Chalamalsetti, S. R., Ndu, G., Foltin, M., Williams, R. S., Faraboschi, P., Hwu, W.-m. W., Strachan, J. P., Roy, K., & Milojicic, D. S. (2019). PUMA: A Programmable Ultra-efficient Memristor-based Accelerator for Machine Learning Inference. *Proceedings of the Twenty-Fourth International Conference on Architectural Support for Programming Languages and Operating Systems (ASPLOS '19)*, 715–731. <https://doi.org/10.1145/3297858.3304049>

[15] Wan, W., Kubendran, R., Schaefer, C. *et al.* A compute-in-memory chip based on resistive random-access memory. *Nature* 608, 504–512 (2022).

[16] Huang, Y., Ando, T., Sebastian, A. *et al.* Memristor-based hardware accelerators for artificial intelligence. *Nat Rev Electr Eng* **1**, 286–299 (2024).

[17] A. Shafiee *et al.*, "ISAAC: A Convolutional Neural Network Accelerator with In-Situ Analog Arithmetic in Crossbars," *2016 ACM/IEEE 43rd Annual International Symposium on Computer Architecture (ISCA)*, Seoul, Korea (South), 2016, pp. 14-26, doi: 10.1109/ISCA.2016.12.

[18] Y. Shuai, N. Du *et al.*, *J. Appl. Phys.* 109, 124117 (2011)

[19] Y. Shuai, N. Du *et al* *Physica Status Solidi(c)*, 10(4): 636-639 (2013)

[20] T. You, N. Du *et al.*, *Adv. Funct. Mater.*, 24,3357 (2014)

[21] N. Du *et al.*, *Front. Neurosci.*, 9, 227 (2015)

[22] X. Zhao *et al.*, "Understanding Stochastic Behavior of Self- Rectifying Memristors for Error-Corrected Physical Unclonable Functions," in *IEEE Transactions on Nanotechnology*, vol. 23, pp. 490-499, 2024, doi: 10.1109/TNANO.2024.3413888.

[23] H. Cai *et al.*, "Realization of Memristor-aided Logic Gates with Analog Memristive Devices," *2022 11th International Conference on Modern Circuits and Systems Technologies (MOCASST)*, Bremen, Germany, 2022, pp. 1-4, doi: 10.1109/MOCASST54814.2022.9837637.

[24] B. Hoffer, V. Rana, S. Menzel, R. Waser and S. Kvatinsky, "Experimental Demonstration of Memristor-Aided Logic (MAGIC) Using Valence Change Memory (VCM)," in *IEEE Transactions on Electron Devices*, vol. 67, no. 8, pp. 3115-3122, Aug. 2020, doi: 10.1109/TED.2020.3001247.

Reviewer #3 (Remarks to the Author):

The paper proposes a novel mixed-mode paradigm for performing logic processing using memristor crossbar arrays. It improves upon state-of-the-art approaches by providing a framework in which both memristance-controlled and voltage-controlled logic processing units can be efficiently integrated, thereby overcoming issues arising out of implementing only one. They do so by proposing extended and improved versions of each and developing a custom automation tool that can synthesize and map arbitrary logic functions into optimal combinations of voltage and memristance-controlled processing units while being aware of physical constraints. Finally, the proposed ideas are validated through experimental demonstrations of two exemplary circuit blocks on memristor crossbar arrays.

While the proposed ideas are novel, comparison with prior work fair and experimental validations impressive, the paper could significantly benefit from better figures (especially the conceptual ones, such that they are easier to interpret by just looking at the figure itself without having to refer to much of the related explanation in main text or caption) and further explanation/elaboration of some of the key concepts/techniques presented. While the current version of the manuscript can already be impactful, resolving the points mentioned below may help make the key concepts and techniques presented more comprehensible and easier for other researchers to reproduce this work.

1. Although the truth tables of the proposed VI³ and MI³ logic processing units are provided there is a paucity of details regarding the standard logic gate emulation capabilities of these proposed units. The manuscript could benefit from elucidation on the logic gates that the MI³ & VI³ can natively implement. This can be done by (a) writing explicitly the Boolean function that these proposed units implement (the Boolean function + digital logic schematic corresponding to their truth tables). Then (b) providing details on how these natively implementable functions can be used to build more complex or a variety of other Boolean functions like AND, OR, XOR, XNOR etc (with 2,3 and more inputs).

Authors' response: Thank you for the reviewer's insightful comments.

To address the comments, we provide below the Boolean expressions that represent the operations of both the MI³ and VI³ logic processing units. These expressions are as follows: the Boolean function for the MI³ unit is represented as

$$m_{y1} = (m_{x1} \cdot \overline{m_{x2}} \cdot \overline{m_{x3}}),$$

while the Boolean function for the VI³ unit is represented as

$$m_{y1} = (\overline{m_{x1}} \cdot v_{x2} \cdot \overline{v_{x3}}) + (m_{x1} \cdot \overline{v_{x2}} \cdot v_{x3}) + (m_{x1} \cdot v_{x2} \cdot \overline{v_{x3}}) + (m_{x1} \cdot v_{x2} \cdot v_{x3}).$$

Further, in response to the reviewer's suggestion to discuss how these functions could be used to build more complex logic gates, we would like to clarify that our approach differs significantly from conventional CMOS-based synthesis methodologies and enables the high parallel in-memory computing. Details are explained as follows: In traditional CMOS design, synthesis tools operate by connecting a range of standard gate types sequentially to achieve the desired output logic, considering only the logical relationships between inputs and outputs. Our M³S automation tool, however, is crossbar-aware, designed specifically for in-memory computing. This approach allows computational results to be directly stored in memory, facilitating immediate access for the next operation within the same cell. In this way, our automation tool is not limited to gate-by-gate construction. Instead, it leverages the unique parallel computing properties of the crossbar

structure, optimizing parallelism for data locality (see example for adder implementation), which is not feasible in a conventional CMOS framework.

Additionally, we would like to emphasize that the benefits and computational efficiency of our approach become more evident when applied to more complex functions. While simpler logic functions, such as basic Boolean gates or even carry-ripple adders, can also be implemented using our method, these examples would not fully capture the advantages of our mixed-mode approach in terms of cycle efficiency and reduced device usage. To effectively demonstrate these benefits, we have chosen to implement computationally intensive functions, such as the 4-bit Sbox designed for use in cryptographic applications. It has been designed on purpose to complicate its algebraic modeling, in order to withstand algebraic attacks against ciphers, so we can consider it among the hardest functions to be handled by algebraic or logical methods, such as our SAT solving. This choice underscores the capabilities of our system in handling complex, real-world computational tasks with optimized performance.

According to the comment of the reviewer, we have added the following discussions in Supplementary Information A (on page 6):

→ The Boolean expressions that represent the operations of both the MI³ and VI³ logic operations are described as follows: the Boolean function for the MI³ operation is

$$m_{y1} = (m_{x1} \cdot \overline{m_{x2}} \cdot \overline{m_{x3}}),$$

while the Boolean function for the VI³ unit is represented as

$$m_{y1} = (\overline{m_{x1}} \cdot v_{x2} \cdot \overline{v_{x3}}) + (m_{x1} \cdot \overline{v_{x2}} \cdot v_{x3}) + (m_{x1} \cdot v_{x2} \cdot \overline{v_{x3}}) + (m_{x1} \cdot v_{x2} \cdot v_{x3}).$$

While simpler logic functions like XOR or XNOR can also be implemented, such examples do not fully demonstrate the strengths of our mixed-mode approach in terms of cycle efficiency and reduced device usage. In this work, we experimentally demonstrate the capabilities of our system by implementing not only a full adder but also computationally intensive functions such as the 4-bit Sbox. The 4-bit Sbox, with its high nonlinearity and resistance to cryptographic attacks, serves as a challenging benchmark to highlight the robust capabilities and optimized performance of our system in managing real-world, complex computational tasks.

2. Although successful experimental demonstration of n-input CRA and 4-bit S-box is shown, it would be great if the steps, techniques and concepts required to map such a circuit using the proposed mixed-mode computing paradigm to the crossbars is elucidated. Authors do provide details related to this in table in Fig. S6 and sub parts of Fig. 3, but the flow is difficult to comprehend. This could potentially be resolved for instance by taking a 1-bit full adder and (a) first describing its schematic using standard logic gates, (b) then its schematic using the natively implementable logic gates provided by the VI³+MI³ hybrid circuit and then (c) schematic of their physical implementation on a crossbar, where each new cycle could be described in a new crossbar diagram. Another important aspect that could be elaborated upon through text/explanation and/or figure/diagram is (d) how would the number of cycles, crosspoint devices required change if the 1-bit full adder is entirely implemented with VI³ or MI³. This would make the message that mixed-mode is better than implementing entirely voltage or memristance-mode more convincing.

Authors' response: Thank you for these valuable suggestions. In response to this suggestion, we have prepared a new diagram Fig. S6 in Supplementary Information E (in the revised manuscript), illustrating 1-bit full adder comparing between schematics using standard logic gates (Fig. S6a) and mixed-mode computing (Fig. S6b). The schematic of their physical implementation on

crossbar (Fig. S6d) is also illustrated for clarifying the mapping process and workflow in mixed-mode computing. Along with it, the implementation flow for 1-bit full adder (Fig. S6c) is summarized, which requires in total 5 cycles, i.e., 3 of VI³ and 2 of MI³ operations.

Despite the newly inserted diagram, in addition, we would like to emphasize that as mentioned in reply to comment 1, scalability and efficiency advantages of mixed-mode computing are better to be demonstrated while handling computationally intensive tasks. Focusing solely on a single 1-bit full adder does not adequately demonstrate the efficiency of the mixed-mode paradigm in a meaningful way. For instance, in the case of the 1-bit ripple carry adder, our mixed-mode approach requires 5 cells and 5 cycles, while a purely in-memory computing approach without reading would require 10 cells and 13 cycles (as referenced in Ref. [17] in Supplementary Information I Table S4). However, the efficiency gains of the mixed-mode paradigm become more apparent as the bit number increases. For instance, implementing an 8-bit full adder in our mixed-mode approach requires only 33 cells and 12 cycles, compared to a purely in-memory computing approach, which would require 87 cells and 97 cycles (also as referenced in Ref. [17] in Supplementary Information I Table S4). This comparison highlights the efficiency of mixed-mode computing in terms of cell and cycle optimization, particularly for multi-bit and complex operations.

a 1-bit full adder based on standard logic gates

b Control sequence of 1-bit full adder

c $a = x_{20}$, $b = x_{30}$, $c = x_{10}$

Cycle	M11	M12	M13	M14	M15
BL ₁ (TE)	x_{10}	x_{10}	x_{10}	x_{10}	x_{10}
1	X	X	X	X	X
WL ₁ (BE)	\bar{x}_{10}	\bar{x}_{10}	\bar{x}_{10}	\bar{x}_{10}	\bar{x}_{10}
BL ₁ (TE)	x_{30}	\bar{x}_{30}	\bar{x}_{30}		x_{30}
2	c_0	c_0	c_0	c_0	c_0
WL ₁ (BE)	\bar{x}_{20}	\bar{x}_{20}	\bar{x}_{20}		\bar{x}_{20}
BL ₁ (TE)		1	0		
3	c_1	v_{40}	v_{70}	c_0	c_1
WL ₁ (BE)		\bar{x}_{30}	\bar{x}_{30}		
BL ₁ (TE)	GND		V_{in}	V_{in}	
4	c_1	v_{50}	v_{80}	c_0	c_1
WL ₁ (BE)					
BL ₁ (TE)	V_{in}	GND		V_{in}	
5	s'_0	v_{50}	v_{80}	c_0	c_1
WL ₁ (BE)					
BL ₁ (TE)		V_r			
Verification	s'_0	s_0	v_{80}	c_0	c_1
WL ₁ (BE)		GND			

d Physical implementation in crossbar

Fig. S6. Implementation of 1-bit full adder using (a) standard logic gates (NOR gate) in comparison to the design using (b) mixed-mode computing. (c) Step-by-step control cycles for applying logic inputs through BLs and WLs in the crossbar configuration. (d) Physical implementation in memristor-based crossbar for each cycle.

According to the comment of the reviewer, we have inserted Fig. S6 into the SI newly and carefully revised the manuscript as follows (on page 14 in Supplementary Information E):

Original text: “For example, as demonstrated in Fig. S7, the 4-CRA can be realized in a 4x5 passive crossbar array with 1R configuration, where no readout is allowed in logic cascading. The computation starts with a VI^3 by programming all cells into a known state c_0 , which in principle allows computation to be started with arbitrary logic inputs.”

Revised text: “For example, as demonstrated in Fig. S6, in the case of the 1-bit carry-ripple adder, each column representing a cell (M11–M14) needed for computation, while M15 stores the output carry bit c_1 . The five columns represent the five cycles required for completing the 1-bit CRA, broken down into three cycles for VI^3 operations and two for MI^3 operations. The VI^3 operations are executed across all cells simultaneously, enabling efficient processing. For each cycle, we specify the WL (Word Line) and BL (Bit Line) inputs used. For implementing 1-bit carry-ripple adder, our mixed-mode approach requires 5 cells and 5 cycles, while a purely in-memory computing approach without reading would require 10 cells and 13 cycles (as referenced in Ref. [17] in Tab. S4 in Supplementary Information I). However, the efficiency gains of the mixed-mode paradigm become more apparent as the bit number increases. For instance, in Fig. S7, the 4-CRA can be realized in a 4x5 passive crossbar array with 1R configuration, where no readout is allowed in logic cascading. Implementing an 8-bit full adder in our mixed-mode approach requires only 33 cells and 12 cycles, compared to a purely in-memory computing approach, which would require 87 cells and 97 cycles (also as referenced in Ref. [17] in Tab. S4 in Supplementary Information I). This comparison highlights the efficiency of mixed-mode computing in terms of cell and cycle optimization, particularly for multi-bit and complex operations.”

3. The SAT formula that is optimized in the mixed mode mapping and synthesis tool could benefit from further elaboration. One suggestion is to take a simple Boolean function (it could be the 1-bit full adder or something simpler) and write down the exact SAT formula (detailing all the clauses) that would be optimized to find the corresponding mixed-mode implementation.

Authors' response:

We thank the reviewer for the insightful suggestion to elaborate on the SAT formula used in the mixed-mode mapping and synthesis tool. Unfortunately, the large number of various constraints and helper variables very quickly makes the Boolean satisfiability formula very long, thus preventing any meaningful description or discussion of it. We generated an instance for the full adder (1-bit CRA). and it had 931 variables and 215,576 clauses in total. We generated minimal examples for circuits consisting just of one AND gate (1 VI^3 operation, no MI^3 operations), which had 45 variables and 933 clauses, and for a circuit of one XOR gate (1 VI^3 operation, 2 MI^3 operations), which had 226 variables and 14,785 clauses. Even these minimal instances are impossible to explain in a paper. Note that our used SAT solving software processes them in a fraction of a second. This may be compared with a numerical solver, where it is possible to write down differential equations being solved, but not to describe in detail the internal representations of the lattice and operations performed.

4. In the first point of the list of limitations that stateful logic faces (in page 4, Section A of SI), authors claim that in such stateful implementations only one type of switching is performed, either SET or RESET. However, even in the proposed MI^3 kernel only a LRS to HRS transition takes place (if any). So, it is not clear to how introducing the MI^3 kernel alleviates this limitation. On the other hand, the VI^3 kernel does allow both types of transition (LRS to HRS as well as HRS to LRS). So, is it through the mixed-mode utilization of MI^3 & VI^3 kernels (not just the sole usage of MI^3) that this issue is alleviated?

Authors' response: Thank you for this insightful observation.

In Supplementary Information A, our goal is to outline the general limitations of MI-based designs, including IMPLY, MAGIC, FELIX, and our MI³ kernel, as shown in Figure S1 (quadrants I and IV). These designs, including MI³, are inherently limited to enabling only one type of transition—typically LRS to HRS—without the flexibility to perform both SET and RESET transitions. Similarly, VI-based designs, discussed in the following paragraphs and represented in quadrants II and III of Figure S1, exhibit their own limitations, such as a reliance on VO readout to achieve universality.

Herewith we would like to emphasize that the goal of our work is not to resolve all limitations of individual kernel but to leverage the complementary strengths of MI³ and VI³ kernels to address the shortcomings of each. By combining the universal capabilities of MI³ with the robustness and bidirectional transition flexibility of VI³ logic designs, the mixed-mode approach achieves enhanced resilience, reduced processing cycles, and the elimination of VO readout, enabling universal and more reliable logic processing.

In order to avoid such misunderstanding, we have carefully revised the manuscript as follows (on page 4 in Supplementary Information A):

Original text: “These issues exacerbate the error rate during cascading of such stateful logic gates. In our study, we introduce the universal MI³ logic operation, which represents an expanded iteration of the MAGIC logic operation. This advancement not only broadens the input variables from 2 to 3, facilitating the cascading process between MI and VI kernels as well as within the MI kernel, but also addresses prevalent challenges such as input drift and partial switching issues within input and output cells (Supplementary Information C), by flexibly combining logic operations in VI kernel before and after MI³.”

Revised text: “These issues exacerbate the error rate during cascading of such stateful logic gates. In our study, we introduce the universal MI³ logic operation, which represents an expanded iteration of the MAGIC logic operation. This advancement not only broadens the input variables from 2 to 3, facilitating the cascading process between MI and VI kernels as well as within the MI kernel (see detailed discussions at the end of this section), but also addresses significant challenges such as input drift and partial switching issues within input and output cells (Supplementary Information C), by flexibly combining logic operations in VI kernel before and after MI³. It is noteworthy that the goal of our work is not to resolve all limitations of individual kernel but to leverage the complementary strengths of MI³ and VI³ kernels to address the shortcomings of each. By combining the universal capabilities of MI³ with the robustness and bidirectional transition flexibility of VI³ logic designs, the mixed-mode approach achieves enhanced resilience, reduced processing cycles, and the elimination of VO readout, enabling universal and more reliable logic processing. “

5. The authors describe the issue of partial switching/compromised state and unwanted drift in both input and output memristances in standard stateful logic implementations and allude to the fact that such issues are reduced by techniques proposed in this work (in Section C of SI). The only thing that does seem to fundamentally reduce such program disturb errors is the idea of using hybrid VI + MI logic units. The maximum error levels caused by cascading pure MI style logic units can be reduced by intermittently introducing VI style logic units in between. However, aside from that, the issue of program disturb still fundamentally exists in the MI³ kernel due to the voltage divider circuit. Authors mention having found an optimal bias of 5.3V that reduces drift in

input memristance. However, finding such optimal biases for voltage divider circuits is an obvious pre-processing step that need to be undertaken for other memristor devices as well. Moreover, such biases are prone to device-to-device variation effects. Therefore, optimal biasing is necessary but not a fundamentally fool-proof way of reducing such drifts. The idea that disturbed states of output memristors in a previous stage of MI³ kernel can be fixed by making it the input memristor in a subsequent stage, does not seem to be intentional, rather a by-product of the cascading process. Wouldn't such a trick be implemented by default when memristance-controlled logic style (either using stateful logic styles in literature like MAGIC or the newly proposed MI³ kernel) are cascaded? Can the authors comment on this? And if it is true, authors should be careful to not imply that such tricks are fundamental and non-trivial contributions of this paper that helps reduce program disturb errors in newly proposed MI³ kernels.

Authors' response: We thank the reviewer for their detailed feedback and appreciate the opportunity to clarify these points. The newly introduced MI³ operation does share some inherent drawbacks of MI-based logic designs, such as IMPLY, MAGIC, and FELIX, including partial switching and drift in both input and output memristances due to the voltage divider circuit. These issues are well-recognized challenges in MI designs. However, our work aims to address these limitations through a co-design strategy that enhances computing accuracy, particularly by leveraging the unique self-rectifying and analog switching characteristics of BFO memristors.

In our previous work (Ref. [1]), we demonstrated how switching dynamics of the underlying memristor technology significantly influence the realization of stateful logic gates like MAGIC NOR. For instance, achieving the MAGIC NOR gate with bipolar memristors often requires a set voltage greater in magnitude than the reset voltage (Ref. [2]). However, with BFO memristors, where the SET and RESET voltage magnitudes are equal, we successfully implemented MAGIC NOR by taking advantage of BFO's self-rectifying behavior. This emphasizes the critical importance of technology-specific optimization in adapting memristor devices for logic designs.

In the current work, we adopt a similar technology-aware strategy tailored to the unique properties of BFO memristors to optimize MI³ operations. Due to BFO's analog switching nature, where current changes gradually instead of abruptly crossing a threshold (as in digital memristors), BFO devices exhibit reduced sensitivity to device variations while maintaining correct logic outputs (large enough memory window between logic '0' and '1'). However, partial switching remains unavoidable in voltage-divider-based logic designs with analog memristors. To ensure stable and accurate outputs, we carefully analyzed the switching kinetics of BFO memristors. These kinetics favor easier transitions from LRS to HRS compared to HRS to LRS. According to this, we structured the MI³ operation to apply positive bias to the top electrodes (TEs) of two parallel-connected memristors while grounding the TE of the third memristor. This configuration provides us the opportunity to exploit cascading for "re-initializing" the logic '1', minimizes the influence of partial switching in the input cells and enhances stability within the cascading MI kernel. Consequently, we designed the MI³ operation to produce a more reliable '0' output by optimizing the bias voltage (V_{in}) to 5.3 V. This bias ensures stable '0' outputs across various input combinations, reducing errors in cascaded operations. It is important to note that this is not a default implementation but a deliberate design choice. For instance, a lower bias of 4.7 V could prioritize precise '1' outputs but at the expense of '0' accuracy, resulting in higher error rates during cascaded operations. Under the current circuit configuration, no re-initialization is possible in cascading steps, further emphasizing the importance of such intentional bias optimization. Additionally, our M³S automation tool is explicitly designed to allow the output cell of MI³ operation can only be reassigned as an output cell in subsequent MI³ operations if it has been used at least once as an input cell in an MI³ or VI³ operation. This

intentional reassignment avoids scenarios where the output '1' from a prior MI³ gate is recomputed into another '1' in the next MI³ cycle, ensures that a more precisely defined input or output value is applied in logic cascading, thereby minimizing error propagation commonly observed in traditional stateful approaches. This approach is not an incidental by-product of cascading but an integral part of our co-design methodology.

To summarize, identifying the optimal bias (5.3 V) for BFO-based voltage divider circuits is indeed an essential preprocessing step and highlights the necessity of technology-aware design. Each memristor technology brings unique characteristics, and our design techniques for BFO memristors, particularly their analog switching behavior, demonstrate how device-specific analysis can maximize computational performance. Other memristor technologies would require similar tailored adaptations.

Moreover, our co-design strategy extends beyond technology-aware optimization. In this work, we intentionally reassign output cells as input cells in subsequent MI³ operations to minimize drift accumulation and maintain higher accuracy in cascaded logic. This, combined with our mixed-mode design incorporating both VI and MI kernels, provides a comprehensive solution to the challenges of memristor-based logic. These optimizations are fully integrated into the M³S automation tool, designed specifically for crossbar architectures. Together, these elements form a holistic co-design framework that is central to our work and demonstrates the necessity of a co-design approach for achieving robust and efficient memristor-based logic computing.

According to the comment of the reviewer, we have carefully revised the manuscript as follows (on page 12 in Supplementary Information C):

Original text: “In mixed-mode computing, we not only exploit variation-resilient VI³ for carrying out partial processing but also design the advanced MI³ to enable robust cascading with multiple MI³ or VI³ operations. To achieve this, we employ $V_{in} = 5.3$ V, completely avoiding the “input drift”—the change of memristance states in input cells (see experimental results in Fig. S3, while ensuring correct transitions from LRS to HRS in the MI³ operation with input combinations '101', '110', and '111' (HRS of M1 after MI³: 111.4 MΩ compared to initialized HRS of M1 at 402.4 MΩ) as demonstrated in Fig. S3. As expected, with $V_{in} = 5.3$ V, the MI³ operation with input combination '100' results in a compromised LRS in M1 after MI³ (LRS of M1 after MI³: 27.2 MΩ compared to initialized LRS of M1 at 1.1 MΩ). Additionally, in the M³S automation tool, we insert conditions to prioritize using the output cell as an input cell in cascaded MI³ (or VI³) operations. Hence, the compromised LRS in M1 can be used as an input cell, where the compromised LRS is refreshed back to logic '1' by the positive bias V_{in} to the TE of M1, while ensuring correct output results, thereby significantly reducing the error rate and enhancing the robustness of MI³ operations during deep cascading. “

Revised text: “Both aforementioned “input drift” and “partial switching” issues are inherent in the stateful logic process (cannot be alleviated). Under mixed-mode computing paradigm, our aim is to mitigate these issues through a co-design strategy that enhances computing accuracy, particularly by leveraging the unique self-rectifying and analog switching characteristics of BFO memristors. This emphasizes the critical importance of technology-specific optimization in adapting memristor devices for logic designs.

Firstly, to ensure stable and accurate outputs, we carefully analyzed the switching kinetics of BFO memristors. These kinetics favor easier transitions from LRS to HRS compared to HRS to LRS. Based on this, we structured the MI³ operation to apply positive bias to the TEs of two parallel-

connected memristors while grounding the TE of the third memristor. This configuration allows us to exploit cascading for “re-initializing” the logic ‘1’, minimizes the influence of partial switching in the input cells, and enhances stability within the cascading MI kernel. Consequently, we designed the MI³ operation to produce a more reliable ‘0’ output by optimizing the bias voltage (V_{in}) to 5.3 V. For instance, as demonstrated in experimental results in Fig. S3, the MI³ operation demonstrates correct transitions from LRS to HRS with input combinations ‘101’, ‘110’, and ‘111’ (HRS of M1 after MI³: 111.4 MΩ compared to initialized HRS of M1 at 402.4 MΩ) as shown in Fig. S3. As expected, with V_{in} = 5.3 V, the MI³ operation with input combination ‘100’ results in a compromised LRS in M1 after MI³ (LRS of M1 after MI³: 27.2 MΩ compared to initialized LRS of M1 at 1.1 MΩ). Thus this bias V_{in} = 5.3 V ensures stable ‘0’ outputs across various input combinations, reducing errors in cascaded operations (see experimental results in Fig. S3). As next step, in the definition of M³S automation tool is explicitly designed to allow the output cell of MI³ operation can only be reassigned as an output cell in subsequent MI³ operations if it has been used at least once as an input cell in an MI³ or VI³ operation. This intentional reassignment avoids scenarios where the output ‘1’ from a prior MI³ gate is recomputed into another ‘1’ in the next MI³ cycle, ensures that a more precisely defined input or output value is applied in logic cascading, thereby minimizing error propagation commonly observed in traditional stateful approaches.”

6. In page 4, Section A of SI, authors claim the following: “This advancement not only broadens the input variables from 2 to 3, facilitating the cascading process between MI and VI kernels as well as within the MI kernel”. It is not clear how and to what extent this advancement is important. Firstly, can the authors quantitatively state the benefits arising out of this additional input in the new 3-input version compared to 2-input versions in literature? Secondly, in what ways is the MI³ kernel responsible for facilitating the cascading between MI and VI kernels? For instance, envision a mixed-mode computing paradigm that comprises of the proposed VI³ kernel (for the voltage-controlled logic unit) and the older 2-input MAGIC-like kernel (for the memristance-controlled logic unit). Wouldn’t a cascading still be possible? If so, then can the authors comment on what is the true benefit of the new MI³ kernel?

Authors’ response: The introduction of a 3-input logic operation does more than simply add an additional input. The advantages can be summarized as follows: Firstly, it embodies a synergistic design that integrates two types of 2-input gates, such as the $\bar{p} \cdot q$ and $\bar{p} + q$ gates from the CRS logic family, into a single 3-input operation. This integrated approach reduces design complexity at the circuit level, enabling faster processing per cycle. The 3-input configuration also allows us to fully exploit both the resistance state and operating voltage as input variables, leveraging the dual physical properties of memristors: stored resistance and applied electrode voltage. By tapping into these two input modes, our design achieves more efficient and accurate logic operations without additional readout logic, leading to improved overall performance.

Secondly, regarding the benefits of the 3-input method for logic cascading, this approach offers substantial efficiency gains over traditional 2-input styles, which require memristive devices to be re-initialized before each operation. In a 2-input system, cascading logic functions would require an additional initialization step because the output from one stage cannot serve directly as input for the next without resetting the device. Our 3-input design, however, enables the output from one operation to be immediately utilized as input in subsequent operations, bypassing the need for re-initialization. This direct integration not only enhances throughput but also reduces the number of sequential stages required, enabling a more streamlined and efficient design flow that supports smooth operation between MI and VI kernels.

Further, the 3-input configuration greatly benefits our automation tool, simplifying complex logic function implementations. Reducing the need for additional stages or complexity in tool design,

this configuration offers a practical solution for efficiently managing memristive systems within a crossbar array, enhancing both automation tool development and overall circuit design.

Finally, we emphasize that this work embodies a holistic co-design approach, integrating device, circuit, and system-level considerations to maximize performance. Addressing each level in isolation would miss out on significant interdependencies that enhance overall system capabilities. In our approach, the 3-input logic design informs and optimizes the automation tool development, while the automation tool, in turn, facilitates effective implementation of the 3-input logic within a crossbar architecture. This coordinated approach enables a high degree of efficiency and flexibility, advancing the system's functionality as a whole.

We believe this response and clarification convey the distinct contributions of our work, which we incorporate into the manuscript as follows for clearer understanding (on page 5 in Supplementary Information A):

Original text: “In this work, we propose a 3-input VI^3 logic operation, representing an extended iteration of CRS logic (without readout). Unlike conventional CRS logic design, this approach expands the scope of input variables from 2 to 3 and establishes a complete gate set through integration with MI^3 logic operation, thereby eliminating the need for a readout operation. This elimination of the readout operation during logic cascading is a significant advantage of our work.”

Revised text: “In this work, we propose a 3-input VI^3 logic operation, representing an extended iteration of CRS logic (without readout). Unlike conventional CRS logic design, this approach expands the scope of input variables from 2 to 3 and establishes a complete gate set through integration with MI^3 logic operation, thereby eliminating the need for a readout operation. This elimination of the readout operation during logic cascading is a significant advantage of our work, which will be detailed further in Supplementary Information C. Besides that, the 3-input logic operations proposed in this work offer several distinct advantages: Firstly, they embody a synergistic design by integrating two types of 2-input gates, such as the $\bar{p} \cdot q$ and $\bar{p} + q$ gates from the CRS logic family, into a single 3-input operation. This integration reduces circuit-level design complexity and enables faster processing per cycle. Additionally, the 3-input configuration fully exploits the dual physical properties of memristors—stored resistance and applied electrode voltage—by using both resistance states and operating voltages as input variables. This approach achieves efficient and accurate logic operations without requiring additional readout logic, significantly improving overall performance. Secondly, the 3-input design provides substantial efficiency gains in logic cascading. Unlike traditional 2-input gates that require memristive devices to be re-initialized before each operation, the 3-input design allows the output from one operation to be immediately reused as input in subsequent operations, bypassing re-initialization steps. This direct integration enhances throughput, reduces the number of sequential stages, and supports a streamlined and efficient design flow, ensuring smooth operation between MI and VI kernels. Furthermore, the 3-input configuration simplifies the implementation of complex logic functions within automation tools by reducing the need for additional stages and design complexity. This makes it a practical solution for managing memristive systems in crossbar arrays, improving both automation tool development and overall circuit design. The last but not the least, our work adopts a holistic co-design methodology that integrates device, circuit, and system-level considerations to maximize performance. By addressing these levels in a coordinated manner, the approach leverages interdependencies that enhance system capabilities. The 3-input logic design optimizes automation tool development, which in turn

facilitates the effective implementation of the design within crossbar architectures, achieving a high degree of efficiency, flexibility, and functionality.”

7. In Section B of SI, the authors claim that VO style kernels can be used to reduce data copying and transmission operations in mixed-mode computation. Authors utilize such kernels only at the beginning of logic processing to facilitate retrieval of input data from memory to input VI kernels. However, will it not be useful also in intermediate stages? For instance, when MI³ kernels are cascaded, it may be possible that the input memristors of a later stage is not physically present at the same crosspoint location as the output memristor of a previous stage that generates that. In such cases, one would have to incur copying/transmission cycles. Can the VO³ kernel not be useful to efficiently relay such physically disjoint memristance values across crossbars?

Authors' response: Thank you for this valuable observation. We agree that VO-style kernels can indeed serve a useful role in intermediate stages, especially in cases where MI³ kernels are cascaded, and input memristors for subsequent stages may not be physically located at the same crosspoint as the output memristors of previous stages. In these situations, the VO kernel would facilitate efficient data transfer across disjoint locations, minimizing the need for additional copying and transmission cycles. In fact, we fully recognize the potential of the VO kernel to relay memristance values across crossbars in such scenarios, and this is one of the key areas we have planned to further explore as a future work. Supplementary Information B demonstrates that when VO kernels are applied, fewer cells and cycles are required to achieve the same logic function, highlighting their utility in reducing overhead.

However, while the VO kernel can be beneficial in intermediate stages, it is still essential to emphasize that the VO kernel is not strictly necessary for all operations and can introduce certain drawbacks, e.g. additional power and latency cost during data transfer between crossbar and crossbar controlling peripherals (see details in Supplementary Information B). Due to these limitations, our focus in this work has been to showcase logic function implementations without using VO kernel during logic processing, which aligns with our goal of optimizing computing performance of logic operations.

To enhance the clarity of these discussions, we have revised our manuscript accordingly as follows (on page 9 in Supplementary Information B):

Original text: “Notably, we intentionally eliminate the utilization of the VO kernel during logical cascading to optimize computational processes. “

Revised text: “Despite in this work we intentionally avoid utilizing the VO kernel during logical cascading to streamline computational processes, it is worthy to mention that VO kernel can play a significant role in intermediate stages, particularly when MI³ kernels are cascaded, and the input memristors for subsequent stages are not physically co-located with the output memristors of prior stages. In such scenarios, the VO kernel can facilitate efficient data transfer between disjoint locations, reducing the need for additional copying and transmission cycles. For instance, the use of the VO kernel during logic processing is demonstrated in Supplementary Information G of the Supporting Information, titled "Proof-of-Principle Demonstration of N-CRA Exploiting VI³ and VO³ Operations." We recognize the potential of the VO³ kernel to relay memristance values effectively across crossbars in these situations, and further exploration of this capability is a key area planned for future work.”

8. Fig. S3 is not easy to read. Please edit the caption and/or figure to make the following fact more easily interpretable: the state S in (a) is the same as m_{x1} in (d-f) and that it indicates the physical configuration of the memristor after a write pulse is applied.

Authors' response: We appreciate the reviewer's suggestion to improve the clarity of Fig. S3. The figure caption has been revised as follows to ensure the information is more easily interpretable (on page 7 in Supplementary Information B):

→ "Figure S3: Demonstration of readout VO^3 logic operations in BFO memristor series. For (a) BFTO, (b) BFO and (c) BiBFO memristive devices, the device schematics, I-V characteristics, logic related switching dynamics, and the state definitions according to the polarity of writing and reading biases are demonstrated (Cylinders: LRS, Cubes: HRS. Red colored shapes represent reconfigurable states in positive bias range, while blue colored ones in negative bias range. Non-reconfigurable states are presented by black colored shapes). The truth tables of technology dependent readout logic VO^3 are determined for (d) BFTO, (e) BFO and (f) BiBFO memristive devices. Note that the state S in (a) corresponds to m_{x1} in (d-f), indicating the physical configuration of the memristor after a write pulse is applied."

9. Type: "demonstrators" to "demonstrations" in first paragraph of page 5 of main text.

Authors' response: We thank the reviewer for pointing out this. We have revised the text accordingly.

10. The authors claim "To the best of our knowledge, this is the first approach to integrated synthesis and mapping of Boolean functions on a crossbar, and also the first to offer optimality guarantees". Is this statement accurate even in light of other works like "SIMPLE MAGIC: Synthesis and In-memory Mapping of Logic Execution for Memristor-Aided logic"?

Authors' response: While previous works, such as *SIMPLE MAGIC: Synthesis and In-memory Mapping of Logic Execution for Memristor-Aided Logic*, have significantly advanced in-memory computing, in comparison to SIMPLE MAGIC work, our approach is innovative in the following aspects: Unlike SIMPLE MAGIC, which relies on classical synthesis tools like ABC and CPLEX, we build directly from Boolean functions, and we employ at least two modes (MI and VI) rather than the single MAGIC logic. Additionally, we integrate synthesis and mapping into a unified process, streamlining computation, whereas SIMPLE MAGIC treats these steps separately.

For instance, SIMPLE MAGIC begins by representing the desired logic function in Berkeley Logic Interchange Format (BLIF) and subsequently converts it into a NOT and NOR netlist through a standard synthesis process. This netlist undergoes further optimization for area (i.e., minimizing the number of gates) using a modified ABC tool. Only afterward is the netlist mapped to the memristor-based memory using a separate mapping strategy with tools like the AMPL modeling language and IBM ILOG CPLEX Optimization Studio, in order to achieve lowest possible latency. This separation of synthesis and mapping, however, introduces inefficiencies, as the mapping process must adapt to a gate netlist that was optimized independently of the crossbar's spatial constraints and parallel computing capabilities.

In contrast, our M^3S approach integrates synthesis and mapping from the outset, simultaneously generating a gate netlist while considering the physical locations of memory cells and the parallelism inherent in crossbar architectures. This co-design methodology ensures that

synthesis directly exploits the crossbar's computing potential, leading to superior efficiency and performance.

Furthermore, our definition of optimality extends beyond minimizing computation steps or cell usage. It includes a holistic co-design framework that unifies for the first time the VI and MI kernels. SIMPLE MAGIC and similar works typically focus on single-kernel designs, such as the MAGIC family based on the MI kernel. By incorporating both VI and MI kernels, our approach explores a broader design space, addressing the limitations of single-kernel methods. This mixed-mode framework reduces cycle counts, improves cell utilization, and minimizes data transfer overheads.

In summary, the integration of mixed-mode design and crossbar-aware synthesis in our method sets it apart from SIMPLE MAGIC and other existing works, supporting our claim to optimality guarantees within this more comprehensive and efficient framework.

To enhance the clarity of these discussions, we have revised our manuscript accordingly as follows (on page 8 in Main Article):

Original text: "To the best of our knowledge, this is the first approach to integrated synthesis and mapping of Boolean functions on a crossbar, and also the first to offer optimality guarantees."

Revised text: "Unlike the previous work [28], which relies on classical synthesis tools like ABC and CPLEX, we build directly from Boolean functions, and we employ at least two modes (MI and VI) rather than the single MAGIC logic (MI mode). Additionally, our approach seamlessly integrates synthesis and physical mapping from the outset, enabling the concurrent generation of a gate netlist while accounting for the spatial distribution of memory cells and the inherent parallelism of crossbar architectures. By adopting this co-design approach, the synthesis process directly leverages the computational capabilities of crossbar systems, ensuring enhanced efficiency, superior performance, and optimality guarantees. "

11. Incorrect sub-figure numbering in Fig. 3 of main text.

Authors' response: We thank the reviewer for pointing out this. We have revised the figure caption accordingly.

Reference list (corresponding to reply to Reviewer 3):

[1] H. Cai *et al.*, "Realization of Memristor-aided Logic Gates with Analog Memristive Devices," *2022 11th International Conference on Modern Circuits and Systems Technologies (MOCAST)*, Bremen, Germany, 2022, pp. 1-4, doi: 10.1109/MOCAST54814.2022.9837637.

[2] B. Hoffer, V. Rana, S. Menzel, R. Waser and S. Kvatinsky, "Experimental Demonstration of Memristor-Aided Logic (MAGIC) Using Valence Change Memory (VCM)," in *IEEE Transactions on Electron Devices*, vol. 67, no. 8, pp. 3115-3122, Aug. 2020, doi: 10.1109/TED.2020.3001247.

Reviewers' comments:

Reviewer #1 (Remarks to the Author):

The authors' response indicates that they may be missing the broader perspective and motivation for their work. While I do not doubt the potential value of in-memory computing with memristors or other emerging memory devices and I agree with the first four sentences in Section H of the Supplementary Information, the discussion in that section and the references cited at the end of the paragraph (Refs. 41-47) do not justify the proposed approach. It is important to highlight a fundamental distinction: Refs. 41–47 report in-memory computing implementations that do not require changing memristors's state each time to compute Boolean logic or dot-product functions. This is a very critical difference and a key point of contention. While I recognize that emerging memory technologies are likely to improve over time, the fundamental challenge remains that write operations in those (nonvolatile, ionic) memory devices are significantly more demanding. Given this, I strongly suggest that the authors include a convincing discussion of at least one (even if it is niche) application that would benefit from the proposed approach. Alternatively, they should explicitly clarify the advances required in memory (memristor) technology to make their approach competitive with conventional technologies. This could be performed for some commonly used operation, e.g., a full adder circuit. Awareness of this fundamental issue is crucial as it would provide valuable insights for device and materials scientists (likely including the authors themselves), enabling them to direct their efforts toward the most impactful advancements.

Authors' response:

We fully understand the concerns from reviewer1 and appreciate the opportunity to further clarify and strengthen the justification for our proposed approach. We would like to address both points mentioned by the reviewer as follows:

Regarding point (1) identifying an application scenario where our mixed-mode approach provides a clear advantage:

We would like to further emphasize that our approach integrates both physical attributes of a memristor device, i.e. **memristance (M) and voltage (V), as active logic variables within a single computational process**, enabling a mixed-mode in-memory computing paradigm that has not been explored in prior works. In contrast to existing in-memory computing methods (e.g. in Ref. [41-47]), which utilize either **memristance (M) or voltage (V)** for logic operations, our approach incorporates both, **providing an additional degree of freedom in computing architectures**. We emphasize that our framework is **not a replacement** for conventional in-memory logic but rather an augmentation that, when combined with prior methods, could **unlock new computational possibilities** while carefully managing switching overhead.

Moreover, importantly, in the manuscript, the VI3 and MI3 logic operations/designs are taken as examples for demonstrating the proposed mixed-mode approach. Meanwhile, we are fully aware of the importance of reducing/minimizing switching effort during computing process. Thus we are actively exploring **readout-based logic designs** that eliminate the need for frequent switching of memristor states. **Scouting Logic design** in published work (<https://ieeexplore.ieee.org/document/7987515>) can be interpreted as one example, which aligns with **MIVO kernel** and avoids switching processes during computation. Figure S1 in the supplementary materials illustrates how such a design could integrate with our proposed framework. Moving forward, we plan to explore and extend our approach by incorporating different **readout-based logic designs** in different computational kernels to minimize the endurance-related limitations.

A possible application domain that can **benefit from our mixed-mode paradigm** is **hyperdimensional computing (HDC)**, which has shown great promise in **machine learning tasks involving temporal patterns**, such as **text classification, biomedical signal processing, multimodal sensor fusion, and distributed sensor networks**. The HDC framework is based on high-dimensional binary vectors (hypervectors) that are processed using operations like **binding, bundling, and permutation** before being stored in associative memory for later retrieval and reasoning. These operations could be performed with our approach directly in-memory. In this case writing-based operations are only used in preparing the associative memory array, which is later used for operation. Thus, the number of writing-based operations are limited allowing for a relaxed endurance requirement. These characteristics make HDC inherently robust to defects, variations, and noise—qualities that align well with **in-memory computing using emerging memristive technologies**.

For instance, in the HDC implementation proposed by **G. Karunaratne** (<https://www.nature.com/articles/s41928-020-0410-3>), memristors are utilized for **in-memory logic operations for binding**, differing from prior approaches that rely on **memristor-based XOR lookup tables** or **digital logic gates**. Given that HDC does not require excessive computational overhead, our **mixed-mode approach—leveraging both voltage and memristance as logic variables—can be seamlessly applied to further enhance computational efficiency**. Additionally, integrating our design with **readout-based kernels** (as discussed earlier) can provide even greater energy savings and endurance benefits, making our approach particularly well-suited for resource-constrained applications.

Regarding point (2) specifying the advances required in memory (memristor) technology to make it competitive with conventional computing:

Here we must consider two critical factors:

- (i) Endurance Considerations: The endurance of memristors is a critical factor in memory technology. While conventional non-volatile memory technologies

require **high endurance ($\geq 10^{16}$ cycles)** for practical development, our approach can mitigate endurance concerns through **distributed and parallel switching strategies**. For instance, if we assume **sequential additions in a memory block**, individual devices would only switch a limited number of times. The endurance requirement can be approximated as:

$$\text{Required endurance} = \frac{\# \text{ switching cycles per device}}{\# \text{ operations performed}}$$

By efficiently distributing operations across memory cells and utilizing techniques such as **write-sharing**, the required endurance per device can be significantly reduced, making the approach more practical.

- (ii) **Energy Efficiency Considerations:** A key metric for competitiveness is that the energy cost of performing operations **must be lower than the energy required for moving data to the CPU**. We estimate energy consumption based on the number of switching events:

$$E_{op} \leq \frac{E_{data\ movement}}{\# \text{ switching events}}$$

Additionally, charging and discharging the crossbar array introduce additional energy overheads, as described in prior studies (e.g., <https://ieeexplore.ieee.org/document/5597913>). In such cases, optimizing the frequency of charge/discharge cycles is crucial to maintaining energy efficiency. Future designs incorporating **smarter charge management strategies** and optimizing peripheral circuitry will further enhance efficiency.

According to the comments of the reviewer, we have carefully included the discussions above in the revised manuscript as follows (on page 22 in Supplementary Information H):

“Building on these advancements, we further emphasize that our approach integrates both memristance (M) and voltage (V) as active logic variables within a single computational process. This mixed-mode operation introduces computing capabilities that were previously unexplored. In contrast to existing in-memory computing methods, which utilize either memristance or voltage for logic computation (e.g., Refs. [41-47]), our approach incorporates both, providing an additional degree of freedom to computing architectures. Rather than replacing conventional in-memory logic, our method augments existing approaches, potentially unlocking new computational possibilities while carefully managing switching overhead.

For instance, our approach can be particularly beneficial for hyperdimensional computing (HDC) [48], which has shown great promise in machine learning tasks such as text classification, biomedical signal processing, and sensor fusion. HDC operations like binding and bundling can be efficiently performed in-memory, with writing-based operations limited to preparing the associative memory array, thereby reducing endurance concerns. As reported in Ref. [48], memristors are utilized for in-memory logic operations for binding. Unlike prior approaches that rely on memristor-based XOR lookup

tables or digital logic gates, our mixed-mode approach leverages both voltage and memristance as logic variables, enhancing computational efficiency.

To make memory technology competitive with conventional technologies, several critical factors must be addressed: 1) **Endurance Considerations**: The endurance of memristors is a critical factor in memory technology. While conventional non-volatile memory technologies require high endurance ($\geq 10^{16}$ cycles) for practical development, our approach can mitigate endurance concerns through distributed and parallel switching strategies. For instance, if we assume sequential additions in a memory block, individual devices would only switch a limited number of times. The endurance requirement can be approximated as:

$$\text{Required endurance} = \frac{\# \text{ switching cycles per device}}{\# \text{ operations performed}}$$

By efficiently distributing operations across memory cells and utilizing techniques such as **write-sharing**, the required endurance per device can be significantly reduced, making the approach more practical. 2) **Energy Efficiency Considerations**: A key metric for competitiveness is that the energy cost of performing operations **must be lower than the energy required for moving data to the CPU**. We estimate energy consumption based on the number of switching events:

$$E_{op} \leq \frac{E_{data\ movement}}{\# \text{ switching events}}$$

Additionally, charging and discharging the crossbar array introduce additional energy overheads, as described in prior studies [48]. In such cases, optimizing the frequency of charge/discharge cycles is crucial to maintaining energy efficiency. Future designs incorporating **smarter charge management strategies** and optimizing peripheral circuitry will further enhance efficiency.

Reviewer #3 (Remarks to the Author):

The authors have made extensive revisions to the entire manuscript based on my comments and I appreciate their detailed responses and clarifications. The revised manuscript now better highlights the key contributions, distinction with prior art, and benefits of the proposed method.

I only two additional comments/suggestions:

- With respect to Table S4, it seems that the area-delay product of the proposed method scales quadratically with the number of inputs in N-bit adder implementations (specifically as $(4N+1) \times (N+4)$), whereas that of Ref 22 (row 5 of Table S4) scales linearly with inputs. From a first look at it, such a quadratic scaling of resources could be quite penalizing for large circuits, especially when such mixed-mode implementations are supposed to be most advantageous for implementing larger/(more complex) circuits.

Authors' response:

We sincerely appreciate the reviewer's constructive feedback and are grateful for the acknowledgment of our revisions. Regarding the concern about the area-delay product scaling in Table S4, we recognize that our proposed method scales quadratically as $(4N+1) \times (N+4)$, whereas the approach in Ref. 22 exhibits linear scaling as $5 \times 15N$. As a direct comparison, it is true that the area-delay product of our approach is more advantageous for small- to medium-bit adders but becomes less favorable beyond 14-bit implementations. However, there are important trade-offs that must be considered: the approach in Ref. 22 does not inherently store the result of the computation, which means that additional storage would be required if result storage is necessary, leading to an area overhead that also scales with N. Additionally, while Ref. 22's approach exhibits better area-delay performance at larger bit widths, it heavily utilizes each device per computation cycle, which significantly impacts endurance—a well-recognized challenge in memristor-based computing. In contrast, our approach distributes computational workload distributed, which may offer advantages in terms of endurance.

According to the comments of the reviewer, we have carefully included the discussions above in the revised manuscript as follows (on page 25 in Supplementary Information I):

“Comparing Ref. [22] with our approach, it is true that the area-delay product of our approach is more advantageous for small- to medium-bit adders but becomes less favorable beyond 14-bit implementations. However, there are important trade-offs that must be considered: the approach in Ref. [22] does not inherently store the result of the computation, which means that additional storage would be required if result storage is necessary, leading to an area overhead that also scales with N.”

Additionally, it is stated in the main paper that the 4-bit S-Box requires 12 memristors whereas the best alternative implementation requires around 100 transistors. While this is roughly an order better in terms of area, it does not say anything about how the area

difference will scale with N for N-bit S-Box nor about the difference in the number of cycles taken (delay). Can the authors comment on this and the overall scaling prospects of their approach for such complex and larger circuits?

Authors' response: We appreciate the reviewer's insightful question regarding the scalability of our approach for larger S-Box implementations. While some S-Boxes follow mathematical constructions that allow scaling for different values of N (where N is a power of 2), cryptographic applications typically optimize S-Boxes for specific use cases rather than arbitrary bit-width scaling. For example, the 4-bit S-Box in MIDORI was determined empirically for energy efficiency, while the 8-bit S-Box in AES follows a structured mathematical design optimized for hardware implementation.

In practice, scaling in cryptographic algorithms occurs not by increasing the bit-width of a single S-Box but by deploying multiple S-Boxes in parallel. For instance, AES uses 16 identical 8-bit S-Boxes to process its 128-bit state, while lightweight block ciphers such as LED use either 16 (for a 64-bit state) or 32 (for a 128-bit state) identical 4-bit S-Boxes. This modular structure implies that any improvements in area or efficiency at the individual S-Box level scale directly to the entire substitution layer, maintaining proportional benefits.

For larger S-Boxes (e.g., 8-bit or 16-bit variants), practical adoption has been limited due to resource constraints, as seen in MK-3, where a 16-bit S-Box required significantly more resources than an AES-style 8-bit S-Box. Given this, the optimal path for scaling remains parallel deployment of smaller S-Boxes rather than increasing their bit-width. The efficiency gains demonstrated at the 4-bit level can therefore be extrapolated to larger cryptographic operations by leveraging their parallel nature.

Regarding area and delay scaling, as mentioned in the manuscript, our approach demonstrates a reduction in device count—requiring only 12 memristors per 4-bit S-Box compared to approximately 100 transistors in conventional designs, yielding an order-of-magnitude area advantage. However, the delay impact depends on the number of computational cycles required for each logic operation. Our implementation benefits from in-memory computing, which can reduce data transfer overheads compared to conventional architectures, though the exact cycle count comparison depends on the logic synthesis of competing approaches.

- In the VI3 row of Table S1, shouldn't the logic inputs be 2 VI (voltage inputs) and 1 MI (resistance input)?

Authors' response:

Thank you for bringing this to our attention. We have revised Table S1 to reflect the correct notation, updating the logic input for VI3 from "MI" to "MI/(VI)" to account for the unknown logic input variables. As illustrated in Fig. S2b, VI3 requires two voltage inputs (VI) and one

memristance input (MI) for computation. The notation adjustment clarifies that while MI is an unknown logic input, VI is placed in parentheses to indicate that voltage inputs can be considered as known or unknown logic inputs depending on the automation design. For example in the proposed design in this work, in order to reduce the data movement between memristive crossbar and peripheral circuits, only known logic values are allowed to be applied as voltage biases to top and bottom electrodes on memristor cell as VI for computation, meaning VI is predetermined (known) rather than unknown during the VI3 logic operation.